# Regenerating hair cells in vestibular sensory epithelia from humans

Ruth Rebecca Taylor[1]*, Anastasia Filia[2], Ursula Paredes[1], Yukako Asai[3], Jeffrey R Holt[3], Michael Lovett[2], Andrew Forge[1]*

[1]UCL Ear Institute, University College London, London, United Kingdom; [2]National Heart and Lung Institute, Imperial College London, London, United Kingdom; [3]F.M. Kirby Neurobiology Center, Boston Children's Hospital, Boston, United States

**Abstract** Human vestibular sensory epithelia in explant culture were incubated in gentamicin to ablate hair cells. Subsequent transduction of supporting cells with *ATOH1* using an Ad-2 viral vector resulted in generation of highly significant numbers of cells expressing the hair cell marker protein myosin VIIa. Cells expressing myosin VIIa were also generated after blocking the Notch signalling pathway with TAPI-1 but less efficiently. Transcriptomic analysis following *ATOH1* transduction confirmed up-regulation of 335 putative hair cell marker genes, including several downstream targets of *ATOH1*. Morphological analysis revealed numerous cells bearing dense clusters of microvilli at the apical surfaces which showed some hair cell-like characteristics confirming a degree of conversion of supporting cells. However, no cells bore organised hair bundles and several expected hair cell markers genes were not expressed suggesting incomplete differentiation. Nevertheless, the results show a potential to induce conversion of supporting cells in the vestibular sensory tissues of humans.
DOI: https://doi.org/10.7554/eLife.34817.001

## Introduction

Loss of the sensory 'hair' cells from the cochlea, the mammalian hearing organ, as a consequence of exposure to ototoxic drugs, excessive noise or through ageing, results in permanent hearing loss. More than 40% of those aged over 50% and 70% of those over 70 have a clinically significant hearing loss (Action on Hearing Loss; www.actiononhearingloss.org.uk/your-hearing/about-deafness-and-hearing-loss/statistics.aspx). Hearing loss has also been reported as a risk factor for dementia (*Livingston et al., 2017*). Loss of hair cells from the vestibular epithelia of the inner ear, results in balance dysfunction causing dizziness and vertigo, significantly under-appreciated disabling conditions. As with hearing loss, the prevalence of vestibular dysfunction increases with age. Dizziness is the most common reason for visits to the GP in those over 75% and 80% of unexplained falls in the elderly are attributable to vestibular dysfunction (*Agrawal et al., 2009*; *Baloh et al., 2001*; *Department of Health, 1999*; *Herdman et al., 2000*; *Pothula et al., 2004*). Regeneration of hair cells could potentially offer a therapeutic approach to amelioriate these conditions.

In the sensory epithelia of the inner ear in all vertebrates each hair cell is surrounded and separated from its neighbours by intervening supporting cells. Hair cells derive their name from the organised bundle of projections from the apical poles. They are mechanotransducers that convert motion into electrical signals. Supporting cells play a role in maintaining the physiological environment necessary for hair cell function and survival, and also repair the lesions in the epithelium when hair cells die. In non-mammalian vertebrates, hair cells lost from the auditory or vestibular sensory epithelia are replaced spontaneously by new ones (*Collado et al., 2008*; *Rubel et al., 2013*). These nascent hair cells are derived from supporting cells. Initially, new hair cells arise from direct, non-mitotic transdifferentiation (phenotypic conversion) of supporting cells into hair cells (*Cafaro et al.,*

*For correspondence:
Ruth.r.taylor@ucl.ac.uk (RRT);
a.forge@ucl.ac.uk (AF)

Competing interests: The authors declare that no competing interests exist.

**eLife digest** The inner ear contains our balance system (the vestibular system) and our hearing organ (the cochlea). Their sensing units, the hair cells, detect movement or sound waves. A loss of hair cells is a major cause of inner ear disorders, such as dizziness, imbalance and deafness.

When hair cells die, supporting cells that surround them close the 'wound' to repair the tissue. In fish, amphibians, reptiles and birds, the supporting cells can replace lost hair cells, but in mammals – including humans – hair cells are unable to regenerate in the cochlea, so hearing loss is permanent. However, previous research has shown that in certain mammals, spontaneous replacement of lost hair cells in the vestibular system can occur, but not enough to lead to a full recovery.

Scientists have been able to convert supporting cells in the vestibular system of mice into hair cells by using either certain chemicals, or by introducing a specific gene into the supporting cells. In the mouse embryo, this gene, called *Atoh1*, switches on a signalling pathway in the inner ear, through which a non-specialised precursor cell becomes a hair cell. Inducing hair cell regeneration could be a therapy for inner ear disorders. Therefore, Taylor et al. wanted to find out if such procedures would work in inner ear tissue from humans.

The researchers collected intact tissue samples from the vestibular system of patients who had undergone surgery to have a tumour removed, which would normally destroy the inner ear. All existing hair cells were removed so that mainly supporting cells remained. Then, the tissue was either treated with chemicals that increased the production of hair cells or received the gene *ATOH1*. The results showed that the cells containing the gene were able to develop many features characteristic of hair cells. And a smaller number of hair cells treated with the chemicals also started to develop hair cell-like features. A gene analysis after the *ATOH1* transfer revealed a number of active genes known to be markers of hair cells, but also several inactive ones. This suggests that additional factors are necessary for generating fully functional hair cells.

Dizziness and balance disorders present a major health care burden, particularly in the elderly population. Yet, they are often disregarded and overlooked. This study suggests that hair cell regeneration could be a feasible therapy for some forms of balance disorders linked to loss of vestibular hair cells. More research is needed to identify the other factors at play to test if hair cell regeneration in the cochlea could be used to treat hearing impairment.

DOI: https://doi.org/10.7554/eLife.34817.002

*2007*; *Taylor and Forge, 2005*). Other supporting cells re-enter the cell cycle, the daughter cells giving rise to hair and supporting cells (*Burns and Stone, 2017*; *Cafaro et al., 2007*; *Collado et al., 2008*; *Rubel et al., 2013*). There is no regeneration in the adult mammalian auditory system. However, there is a limited capacity to regenerate hair cells in vivo in the mammalian vestibular system (*Forge et al., 1993*; *1998*; *Lopez et al., 1997*; *Kawamoto et al., 2009*). These hair cells arise by direct phenotypic conversion of supporting cells (*Li and Forge, 1997*; *Lin et al., 2011*). We recently reported the presence of cells bearing immature hair bundles in the vestibular system of elderly people (*Taylor et al., 2015*). This suggests that a capacity to regenerate hair cells may exist at a very low level throughout life in humans.

During development, supporting cells and hair cells are derived from the same homogeneous population of precursor cells following a terminal mitotic event (*Kelley, 2006*). The mosaic patterning of hair and supporting cells develops by lateral inhibition mediated by the Notch signalling pathway (*Kiernan, 2013*). In cells differentiating as hair cells the basic helix-loop-helix transcription factor Atonal homolog 1 (Atoh1) is transiently expressed and has been shown to be necessary for sensory precursors to differentiate into hair cells (*Bermingham et al., 1999*; *Kelley, 2006*; *Woods et al., 2004*). The nascent hair cells express the Notch ligand Delta1 on their surface which activates the Notch receptor, a transmembrane protein, in adjacent neighbouring cells. The inhibitory activity of Notch prevents the adjacent cell following the same fate, so this cell will not become a hair cell but instead will be a supporting cell. Ligand binding to the Notch receptor triggers the extracellular cleavage of Notch by tumour necrosis factor alpha converting enzyme (TACE) (*Kopan and Ilagan, 2009*) and intracellular cleavage by γ-secretase (*Kiernan, 2013*). This releases the Notch intracellular domain which enters the nucleus interacting with several transcription factors to suppress *Atoh1*

expression thereby inhibiting differentiation as a hair cell and promoting HES/HEY expression, propelling those cells to become supporting cells.

Atoh1 is expressed during hair cell regeneration in chick and zebrafish and several studies have shown that ectopic overexpression of Atoh1 in the organ of Corti or vestibular sensory epithelia of mammals is sufficient to induce generation of cells that express hair cell marker proteins (*Zheng and Gao, 2000*). These arise by direct transdifferentiation without an intervening mitotic event. Overexpressing *Atoh1* or using γ- secretase inhibitors, potentially offer means to induce supporting cells to become hair cells. Both these approaches have been applied to murine vestibular epithelia depleted of hair cells (*Lin et al., 2011*; *Staecker et al., 2007*).

We have established a consortium of surgeons throughout the UK to harvest human vestibular epithelia from translabyrinthine operations for the removal of acoustic neuromas (vestibular schwannomas) (*Taylor et al., 2015*). Here we use the vestibular epithelium collected from such surgeries to examine the capacity of supporting cells to generate new hair cells in adult human inner ear tissue. We have used an adenoviral vector to deliver *ATOH1* (adV2- ATOH) to transduce cells in human sensory epithelia from which hair cells have been ablated with the ototoxic agent, gentamicin. We find that significant numbers of cells expressing hair cell markers can be generated. We have also exposed tissue to the γ- secretase inhibitor TAPI1 (TNFα protease inhibitor 1), a TACE inhibitor. Cells expressing hair cell marker proteins are also generated but in fewer numbers that with *ATOH1* transduction. However, neither protocol resulted in fully differentiated hair cells. Studies of gene expression following ATOH1 transduction by RNA sequencing confirmed that a significant cascade of downstream effectors is induced by this treatment. However, this induction falls short of complete hair cell conversion but highlights components that may be necessary in completing the conversion events.

## Results

Utricles harvested from vestibular schwannoma surgeries were collected in medium and transported to the laboratory as previously described (*Taylor et al., 2015*). Tissue samples were examined and assessed before incubation in medium for ca. 18 hr prior to exposure to gentamicin or as a control in medium alone. Samples were used only when an intact epithelium was clearly visible under microscopic examination. Several harvested samples were not suitable for experimental use being damaged on excision or transfer. The age of the patients ranged between 17 and 81 (mean 50.6, median 51) but there was no selection or exclusion of samples on the basis of age or gender of the donor.

### Untreated utricles

In utricles fixed and processed immediately following harvesting, hair cells, labelled for myosin VIIa, a hair cell marker, were present across the entire epithelium (*Figure 1A*), but, as reported previously (*Taylor et al., 2015*), the density of hair cells varied and in some samples, particularly those from older individuals, there were very few. SEM showed the characteristic hair bundles of vestibular hair cells with stereocilia of graded height but there was considerable variability in the morphology of the bundles and the height of the longest stereocilia in each bundle (*Figure 1B*). Short microvilli covered the apical surfaces of supporting cells. Thin sections of untreated utricles revealed a bilayer of cells with rounded hair cell nuclei in the more apical region closer to the luminal surface, while the more irregularly shaped supporting cell nuclei were located close to the underlying basement membrane (*Figure 1C,D*). Hair cells survived in vitro in most untreated samples maintained in explant culture for 28 days although in some cultures incubated for this period, condensed, mis-shapen remnants also labelled positively for myosin VIIa suggesting a possible incipient deterioration in the cultures by this time (*Figure 1D*). This defined a period of 21–22 days for an optimal total time of incubation in subsequent experiments, a period of sufficient length to cover that over which spontaneous regeneration of hair cells occurs in the vestibular organs in vivo in guinea pigs (*Forge et al., 1993*, *1998*), chinchillas (*Lopez et al., 1997*) and mice (*Kawamoto et al., 2009*).

### Gentamicin treatment results in extensive hair cell loss

To ablate hair cells we exposed utricles to the ototoxic aminoglycoside antibiotic gentamicin. At 24 hr following 48 hr exposure to 2 mM gentamicin, few intact hair cells remained. Apoptotic death of hair cells in the body of the epithelium was evident by positive immunolabelling for activated

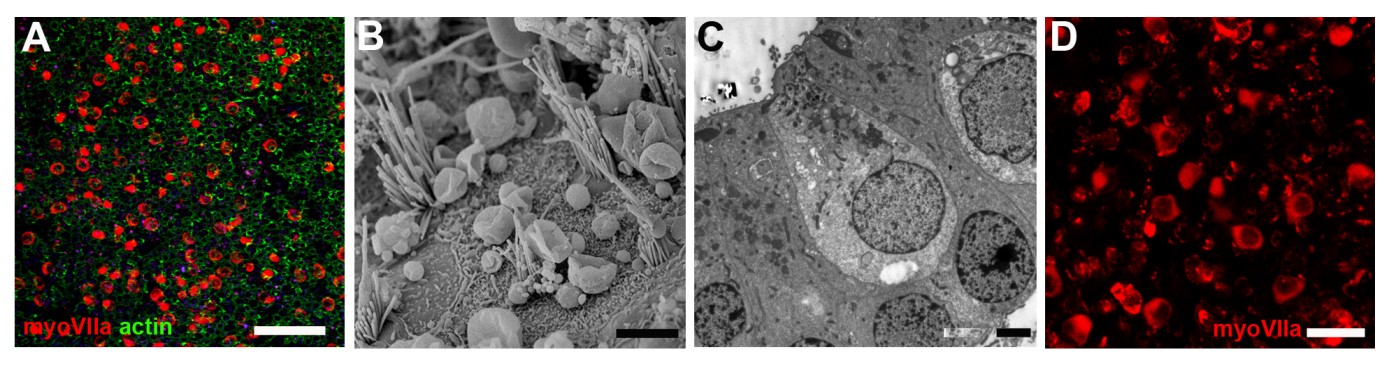

**Figure 1.** Undamaged utricular maculae. (A–C) Fixed immediately after harvesting. (D) After 28 days in explant culture. (A) Hair cells labelled for myosin VIIa (red) are distributed over the entire utricle. Phalloidin labels actin (green) at the intercellular junctions at the luminal surface of the epithelium. Many of the junction-associated actin bands are quite wide, but others, mainly where hair cells have been lost, are thin. Scale bar: 50 μm. (B) SEM shows organised hair bundles of hair cells and surfaces of intervening supporting cells. Scale bar: 5 μm. (C) Hair cells do not contact the basement membrane underlying the epithelium and their nuclei are at a level above that of supporting cells, which are in contact with the basement membrane. Neuronal elements have rapidly died away from the hair cells. Scale bars: 5 μm. (D) Undamaged utricle maintained in explant culture for 28 days. Hair cells labelled for myosin VIIa (red) are distributed across the entire epithelium. Scale bar: 25 μm.

DOI: https://doi.org/10.7554/eLife.34817.003

caspase 3 (not shown) and in thin sections for TEM by pyknotic nuclei or marginated chromatin, with apoptotic bodies inside supporting cells (*Figure 2A,B*). Some hair cells were also seen to be extruded from the epithelium (*Figure 2C*). Both apoptosis, with apoptotic bodies phagocytosed by supporting cells, and extrusion of hair cells from the epithelium have been observed to occur in the vestibular sensory epithelia in vivo in animals treated with aminoglycoside (*Li et al., 1995*). With loss of hair cells, supporting cells closed the lesions (*Figure 2D*). The number of hair cells in cultures treated with gentamicin (N = 16) was assessed by counting myoVIIa positive cells in two groups: an early timepoint, 2–4 days post-gentamicin (dpg) (N = 5); and a late timepoint, 11–21 dpg (N = 11). At the earlier stage there was a mean of $5.26 \pm 1.48$ per 10000 $\mu m^2$. The hair cell bodies that persisted in the first few days after gentamicin exposure were scattered across the epithelium (*Figure 2E*). They were always rounded in shape and most contained an actin-rich rod-like inclusion structure (*Figure 2E,F,G*) indicative of pathology; they are reminiscent of the 'cytocaud' observed in damaged guinea pig vestibular hair cells (*Kanzaki et al., 2002*), and in mice appear in damaged hair cells destined for phagocytosis by supporting cells (*Bucks et al., 2017*). In tissue cultured for longer periods, up to 21 days post gentamicin treatment, there were very few hair cells (*Figure 2H*, ca. $1.4 \pm 0.31$ per 10000 $\mu m^2$). The surfaces of almost all cells across the epithelium were of similar appearance (*Figure 2I*), with no surface projections or other structural specialisations, except for dispersed short microvilli, and with a polygonal outline, features characteristic of supporting cells following loss of hair cells. There was no evidence of spontaneous regeneration of hair cells. Thus, the prolonged high dose exposure to gentamicin ablated the majority of hair cells resulting in an epithelium composed predominantly of supporting cells.

## Adenoviral transduction of human vestibular epithelium

To promote hair cell generation by expression of exogenous *ATOH1* transduction we used a second generation multiply-depleted replication-incompetent adenoviral vector, Ad2, previously shown to transduce human hair cells and supporting cells in vitro (*Kesser et al., 2007*). The vector carried the genes for green fluorescent protein (GFP) and atonal homologue 1 (ATOH1) independently driven by CMV promoters. After ablation of hair cells by incubation with gentamicin, utricles (and some cristae) were thoroughly rinsed with medium and incubated for up to 24 hr with the viral vector in serum-free medium. Expression of GFP at 4 days after transduction showed that supporting cells in the human tissue could be efficiently transduced, and delineated variable shapes of supporting cells revealing some with thin basally directed projections (*Figure 3A*). Labelling for SOX2, which is

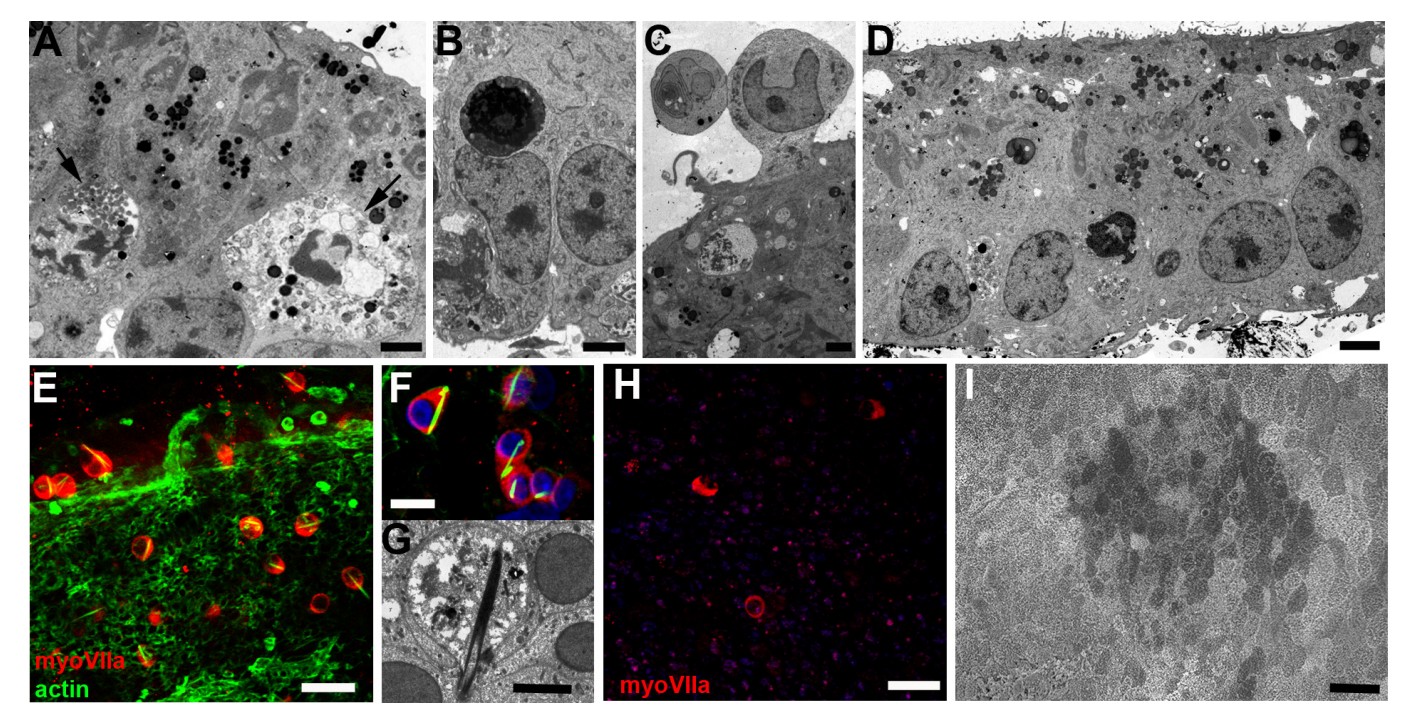

**Figure 2.** Hair cell loss in utricular maculae following incubation in gentamicin. (A) Hair cells undergoing apoptosis (arrows), indicated by the condensed marginated chromatin in their nuclei, at 24 hr after incubation in gentamicin. Scale bar: 2 µm. (B) Apoptotic body inside a supporting cell at 24 hr after incubation with gentamicin. Scale bar: 2 µm. (C) Hair cells in the process of extrusion from the sensory epithelium at 24 hr after incubation in gentamicin. Supporting cells below the extruding hair cell have formed tight junctions to close the space that the hair cell is vacating. Scale bar: 2 µm. (D) Four days after the end of incubation with gentamicin, few hair cell bodies are evident within the epithelium. Scale bar: 5 µm. (E) 2 days after incubation with gentamicin, remaining hair cells labelled for myosin VIIa (red) are almost all rounded in shape and enclose a phalloidin-labelled rod-like inclusion (green). Scale bar: 25 µm. (F) Rod-like inclusions composed of actin (labelled with phalloidin, green) are present in almost all remaining hair cells (labelled for myosin VIIa, red) at 24 hr after incubation with gentamicin. Nuclei labelled with DAPI (blue). Scale bar: 10 µm. (G) Thin section reveals rod-like inclusion composed of densely packed filaments in degenerating hair cell at 24 hr after incubation with gentamicin. Scale bar: 2 µm. (H) 21 days after incubation with gentamicin very few cells that label for myosin VIIa remain. Scale bar: 25 µm. (I) SEM of apical surface of utricle 21 days after incubation with gentamicin. There are no hair bundles or other obvious surface specialisations. The surfaces of almost all cells are relatively smooth, with variable numbers of short dispersed microvilli, and have a polygonal profile, characteristics of the apical surfaces of supporting cells in regions where hair cells have been lost. Scale bar: 20 µm.

DOI: https://doi.org/10.7554/eLife.34817.004

expressed by supporting cells, showed that GFP was co-expressed in cells with nuclei positively labelled for SOX2 (*Figure 3B*). Expression of ATOH1 could be detected by immunolabelling in some cells expressing GFP both in utricles (*Figure 3C*) and in cristae (*Figure 3D*), but not all GFP expressing cells also expressed ATOH1. Likewise, myosin VIIa was also expressed in some cells expressing GFP but not others (*Figure 3E*). By 17 days after incubation with the virus (tissue cultured for a total of 20 days) many myosin VIIa-positive cells were apparent (*Figure 3F*). These myosin VIIA-positive cells were often tightly packed together, contacting each other in some regions of the tissue (*Figure 3G*) and of variable shapes, rarely rounded but often elongated and some with thin basally directed projections (*Figure 3H*.) Only some cells expressed both GFP and myosin VIIA (*Figure 3G, H*) but in many that did, GFP was less strongly expressed than in cells expressing GFP alone (*Figure 3G*), perhaps suggesting that GFP may be downregulated over time as cells differentiate. The number of cells expressing myoVIIa (14.54 ± 3.68, N = 7) was significantly greater than the number in control tissue from which hair cells had been ablated and then maintained for an equivalent period of time without further treatment (*Figure 4*).

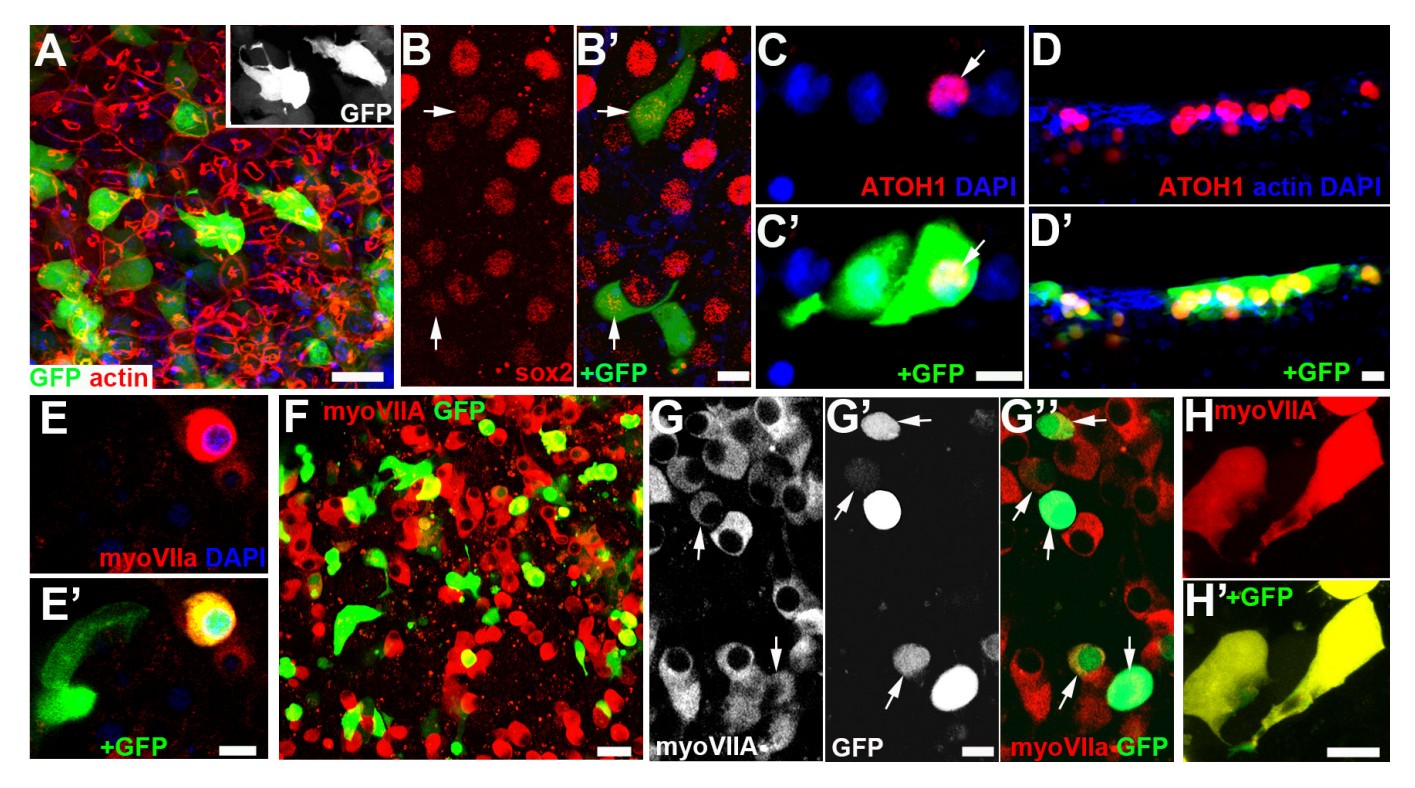

**Figure 3.** Sensory epithelia transduced with *ATOH1* and *GFP* after incubation with gentamicin. (**A**) GFP expression 2 days after transduction. Many supporting cells express GFP which delineates cell shape revealing several cells with thin, basally directed extensions (inset). Scale bar 20 μm. (**B**) Labelling for SOX2 (red), and **B'** SOX2 and GFP expression, 2 days after transduction. Arrows indicate nuclei labelled for SOX2 (**B**) inside cells that express GFP (**B'**). Scale bar: 10 μm. (**C**) Frozen section. Labelling for ATOH1(red) and **C'** ATOH1 with GFP expression in utricular macula 5 days after transduction. Arrows indicate nucleus labelled for ATOH1 (**C**) in cell that expresses GFP (**C'**). Adjacent cell, with basally directed extension expresses GFP but the nucleus does not label for ATOH1. Scale bar: 10 μm. (**D**) Frozen section. Labelling for ATOH1(red) and **D'** ATOH1 with GFP expression in crista 5 days after transduction. Several cells whose nuclei label for ATOH1 also express GFP, but there are also cells with ATOH1 +nuclei which do not express GFP. Phalloidin-labelled actin, as well as DAPI to label nuclei, is in the blue channel to label the intercellular junctions as orientation for identification of the luminal surface of the epithelium. Scale bar: 10 μm. (**E**) Myosin VIIa (red) labels cell that also expresses GFP (**E'**). Adjacent cell, with morphological characteristics of supporting cell, expresses GFP but not Myosin VIIa. Scale bar: 10 μm. (**F**) 18 days after transduction, many cells label for Myosin VIIa (red). Scale Bar: 20 μm. (**G**) Myosin VIIa labelling, (**G'**) GFP expression, (**G''**) Merge of myosin VIIA labelling (red) and GFP at 18 days post transduction. Arrows indicate some cells expressing GFP that label for myosin VIIa. The intensity of labelling for myosin VIIa and the level of expression of GFP vary. Myosin VIIa labelled cells often clustered together and appear to be in contact. Scale bar: 10 μm. (**H, H'**) Some cells at 18 days after transduction that are labelled for myosin VIIa (**H**) and also express GFP (**H'**) have thin basally extended projections. Scale bar: 10 μm.

DOI: https://doi.org/10.7554/eLife.34817.005

## Inhibition of the notch signalling pathway

Studies in birds and mammals have shown that replacement hair cells can be derived from the supporting cells that remain after hair cell loss through inhibition of the Notch-signalling pathway (*Daudet et al., 2009*; *Lin et al., 2011*; *Warchol et al., 2017*). To test the hypothesis that hair cell regeneration in human inner ear tissue could be induced by inhibition of Notch signalling, we maintained gentamicin treated cultures (N = 7) in TAPI-1 for 18 days. TAPI-1 is a potent small molecule inhibitor of matrix metalloproteinases and TACE (TNF-α convertase/ADAM17). Myosin VIIA+ cells were quantified and numbers compared with damaged utricles cultured for a similar duration (3.85 ± 1.2 vs 1.39 ± 0.3, respectively; *Figure 4*). Statistical analyses revealed a significant difference between gentamicin-only treated tissue and samples subsequently maintained with TAPI-1 ($p < 0.05$) but also that the number of myosinVIIA positive cells following incubation with TAPI1 was significantly less than that generated by ATOH1 transduction.

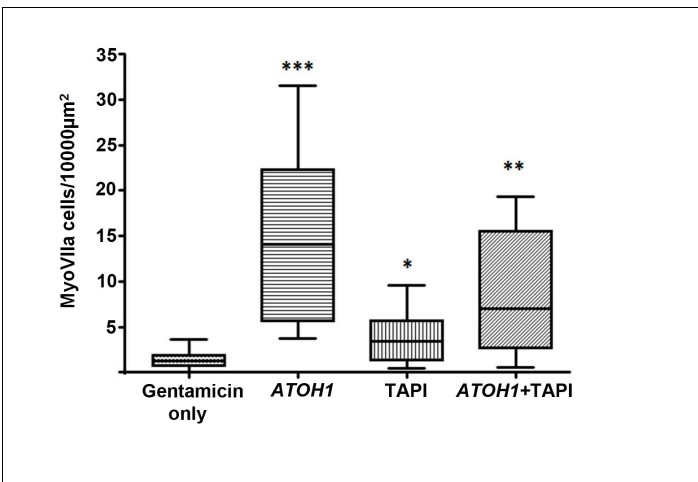

**Figure 4.** Numbers of myosin VIIa positive cells per unit area. Number of labelled cells (25%–75% percentiles and maximum-minimum) in utricles following 48 hr incubation in gentamicin then maintained up to 28 days with no further treatment (Gentamicin only, control tissue N = 11); 18 days after transduction with *ATOH1* (N = 7); maintained 18 days in medium containing TAPI1 (N = 7); transduced with *ATOH1* and maintained 18 days in medium containing TAPI1 (N = 6). Asterisks indicate significant difference in the numbers of myosin VIIa positive cells in each treatment condition compared to the gentamicin only control.
DOI: https://doi.org/10.7554/eLife.34817.006

To test whether we could further enhance the generation of 'hair cells' seen in tissue transduced with ATOH1, we exposed damaged utricles (N = 6) to Ad2-ATOH1-GFP and then maintained them in medium with TAPI-1. Unexpectedly, the number of myoVIIA+ cells was lower than tissue transduced with Ad2-ATOH1-GFP and maintained in medium alone (8.39 ± 02.65). However, there were more myosin VIIA+ cells than found in damaged cultures exposed to TAPI-1 alone. Overall, the number of myoVIIa positive cells in all three treatment groups (ATOH1 transduced; TAPI-1 alone; and ATOH1 transduced +TAPI-1) was statistically significantly greater than that in gentamicin-only treated utricles (*Figure 4*) providing evidence that supporting cells can be manipulated to generate hair cell-like cells in human inner ear sensory epithelia. There was no significant difference in the mean ages of the donors in each treatment group and the age ranges were similar (control: mean 58.1, range 36–81; ATOH1 transduced: mean 55.7, range 41–67; TAPI1: mean 46.8, range 32–64; ATOH1 +TAPI: mean 46.7; range 20–71). This indicated that age is unlikely to be a factor underlying the difference between treatment regimes in the number of myosin VIIa cells generated.

## Morphological assessment

In explants exposed to ATOH1, either with or without TAPI1, following the ablation of hair cells with gentamicin, SEM (3 samples of each condition) showed the apical surface of many cells across the epithelium had bushy microvillar projections that were noticeably longer than the microvilli of neighbouring cells (*Figure 5A*). Some cells had a central protrusion within the cluster of microvilli, reminiscent of the position of the kinocilium seen in immature hair bundles during development of the sensory epithelia and hair cell regeneration (*Denman-Johnson and Forge, 1999*; *Forge et al., 1998*; *Tilney et al., 1992a*; *Tilney et al., 1992b*) (*Figure 5B,C*). The projections were present on cells that expressed GFP and labeled for myosin VIIa and were composed of actin (*Figure 5D*), but no cells showed organized hair bundles, and in whole mount samples prepared for immunolabelling, very little labelling for espin, a known actin bundling protein expressed along the length of stereocilia (*Zheng et al., 2000*), could be detected. In thin sections, cells with prominent elongated microvilli were identified. They were also evident in samples that had first been examined by SEM (two samples) (*Figure 5D,E*) as well in samples which had been embedded after immunolabelling for myosin VIIa (two samples) (*Figure 5F,G*). These cells showed some features resembling hair cells: they were generally cylindrical in shape with a rounded nucleus located towards the luminal pole and, when their profiles were followed through the serial sections, they were not in contact with the basement

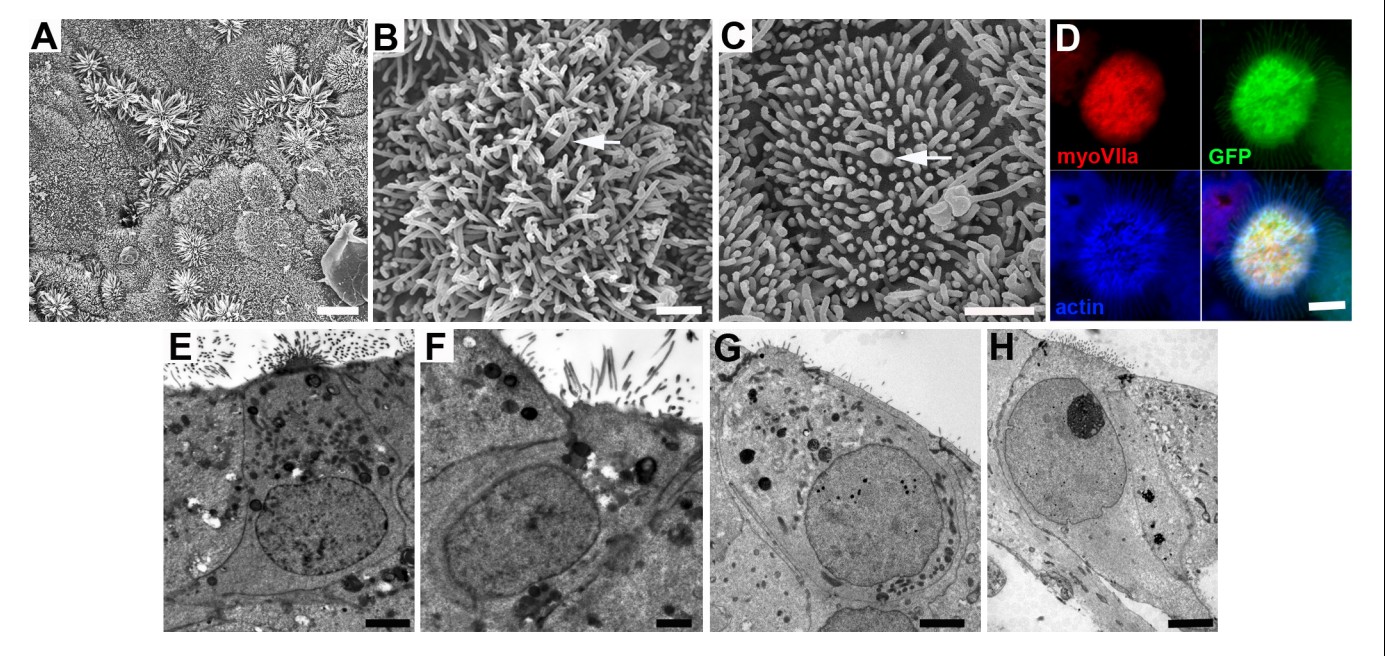

**Figure 5.** Morphological characteristics of cells in utricles 18 days after transduction with *ATOH1*. (A) Many cells across the epithelium bear elongated microvillar-like projections from their apical surface. Scale bar: 5 µm. (B, C) Cells with dense cluster of apical projections possess a single thicker, kinocilium-like projection from the centre of the cell surface (arrowed). Scale bars: 1 µm. (D) Thin projections on cells expressing myosin VIIa (red) and GFP contain actin (blue). Scale bar: 2 µm. (E,F) Thin sections of cells in utricle in which SEM showed cells with dense clusters of apical projections in panel A, and (G,H) in a utricle that showed large numbers of myosin VIIa labelled cells after immunolabelling similar to the utricle shown in *Figure 3* panel F. In cells with numerous, microvillar projections, nuclei are located towards the apical (luminal) surface and have approximately cylindrical cell bodies, similar to hair cells, but some (panels E and H) have thinning basal extensions, reminiscent of supporting cells converting to hair cells. Scale Bars: 2 µm.

DOI: https://doi.org/10.7554/eLife.34817.007

membrane, although some cells had thinner elongated basal projections similar to that seen with myosin VIIA labelling and expected of a cell converting from a supporting cell (*Li and Forge, 1997*; *Taylor and Forge, 2005*). However, these cells did not possess cuticular plates - the actin meshwork that forms a platform beneath the stereocilia in differentiating hair cells - nor were there any synaptic specializations such as synaptic ribbons (*Taylor et al., 2015*). Thus, it appeared that while cells expressing hair cells markers could be generated, those cells did not differentiate fully as hair cells.

## Gene expression changes with Ad2-ATOH1-GFP transduction

To investigate alterations in gene expression after Ad2-ATOH1-GFP transduction of human supporting cells, we measured transcriptome changes by RNA-seq. We compared gentamicin-treated tissue transduced with Ad2-ATOH1-GFP cultured for 5 days, versus control tissues only treated with gentamicin (cultured for 2 days, 8 days and 14–18 days). Three separate comparisons were performed between the Ad2-ATOH1-GFP transduced tissues and the three different controls to limit the variability between control samples and derive a consistent set of gene expression changes. Based upon this stringent comparison, a total of 494 genes were significantly differentially expressed ($-2 >=$ fold change $>= 2$ and $p < 0.05$ across all three comparisons). The expression of 441 out of these 494 genes exhibited the same direction of change in all three comparisons. Of these, 53 were down-regulated by ATOH1 transduction whereas 388 were upregulated. ATOH1 was among the most significantly overexpressed genes in tissues transduced with ATOH1 (70 to 286-fold $< p < 0.05$).

Within the 441 significantly differentially expressed genes we were interested in identifying key regulatory genes that showed consistent changes in gene expression. Including ATOH1, a total of 18 transcription factors (TFs) and five known chromatin modifiers (CMs, one gene SATB2 has both

**Table 1.** Transcription factors and chromatin modifiers that exhibit significant changes in gene expression upon Atoh1 transduction into human sensory epithelia.

The left column lists gene names followed by gene descriptions. AB T is the average transcript abundance in the Atoh1 treated samples (as FPKMs). AB C is the average abundance in the control samples. LOG2 is the log base2 fold change of Atoh1 transfected compared to the controls. FUNCT indicates whether a given gene is a transcription factor (TF) or a chromatin modifier (CM). Names in bold are genes that show a > 2 fold change in expression (and p<0.05) across all three comparisons (they are part of the 441 differentially expressed genes, see Materials and methods and Results). Genes in plain text show statistically significant changes in all comparisons, but only pass >2 fold in at least one comparison.

| GENE | Description | Ab T | Ab C | LOG2 | Funct |
|---|---|---|---|---|---|
| ATOH1 | atonal bHLH transcription factor 1 | 37.8 | 0.3 | 6.9 | TF |
| RAX | retina and anterior neural fold homeobox | 0.7 | 0.1 | 2.5 | TF |
| ZNF296 | zinc finger protein 296 | 17.1 | 3.0 | 2.5 | TF |
| POU4F3 | POU class 4 homeobox 3 | 3.4 | 0.8 | 2.1 | TF |
| CITED4 | Cbp/p300 interacting transactivator | 56.6 | 13.7 | 2.0 | TF |
| GTF2IRD2 | GTF2I repeat domain containing 2 | 2.7 | 0.7 | 2.0 | TF |
| SNAI1 | snail family transcriptional repressor 1 | 30.7 | 8.9 | 1.8 | TF |
| ZNF775 | zinc finger protein 775 | 17.8 | 5.9 | 1.6 | TF |
| MEF2B | myocyte enhancer factor 2B | 2.1 | 0.7 | 1.6 | TF |
| SOX12 | SRY-box 12 | 10.1 | 3.6 | 1.5 | TF |
| ZNF784 | zinc finger protein 784 | 3.7 | 1.3 | 1.4 | TF |
| ZNF837 | zinc finger protein 837 | 1.8 | 0.7 | 1.4 | TF |
| IRF9 | interferon regulatory factor 9 | 137.3 | 53.5 | 1.4 | TF |
| ZNF692 | zinc finger protein 692 | 36.7 | 15.1 | 1.3 | TF |
| GTF2H3 | general transcription factor IIH subunit 3 | 9.3 | 21.5 | −1.2 | TF |
| ZNF382 | zinc finger protein 382 | 0.9 | 2.2 | −1.2 | TF |
| EP300 | E1A binding protein p300 | 10.9 | 26.3 | −1.3 | CM |
| ZFHX4 | zinc finger homeobox 4 | 1.0 | 2.4 | −1.3 | TF |
| ASH1L | ASH1 like histone lysine methyltransfer | 5.1 | 12.8 | −1.3 | CM |
| ATRX | ATRX, chromatin remodeler | 7.0 | 22.4 | −1.7 | CM |
| SATB2 | SATB homeobox 2 | 0.8 | 3.0 | −1.9 | TF/CM |
| HDAC9 | histone deacetylase 9 | 5.7 | 20.9 | −1.9 | CM |
| TCF19 | transcription factor 19 | 0.7 | 0.2 | 1.8 | TF |
| HES1 | hes family bHLH transcription factor 1 | 78.1 | 24.1 | 1.7 | TF |
| ZNF467 | zinc finger protein 467 | 4.3 | 1.5 | 1.5 | TF |
| GSC | goosecoid homeobox | 0.8 | 0.3 | 1.5 | TF |
| SALL1 | spalt like transcription factor 1 | 9.7 | 3.9 | 1.3 | TF |
| KCNIP3 | potassium voltage-gated channel protein 3 | 10.1 | 4.3 | 1.3 | TF |
| THAP3 | THAP domain containing 3 | 30.6 | 13.2 | 1.2 | TF |
| MSX1 | msh homeobox 1 | 8.4 | 3.7 | 1.2 | TF |
| SCAND1 | SCAN domain containing 1 | 58.9 | 26.8 | 1.1 | TF |
| ING2 | inhibitor of growth family member 2 | 13.2 | 6.1 | 1.1 | CM |
| MLXIP | MLX interacting protein | 8.8 | 4.2 | 1.1 | TF |
| ZNF652 | zinc finger protein 652 | 11.4 | 5.7 | 1.0 | TF |
| PATZ1 | POZ/BTB and AT hook containing zincfinger | 18.1 | 9.3 | 1.0 | TF/CM |
| RARG | retinoic acid receptor gamma | 18.0 | 9.2 | 1.0 | TF |
| GLI4 | GLI family zinc finger 4 | 13.1 | 6.8 | 1.0 | TF |
| THAP8 | THAP domain containing 8 | 7.5 | 4.3 | 0.8 | TF |

*Table 1 continued*

| GENE | Description | Ab T | Ab C | LOG2 | Funct |
|------|-------------|------|------|------|-------|
| ING5 | inhibitor of growth family member 5 | 17.5 | 10.4 | 0.8 | CM |
| TAF6 | TATA-box binding protein associated factor | 42.6 | 26.1 | 0.7 | TF |
| ZNF768 | zinc finger protein 768 | 47.2 | 29.3 | 0.7 | TF |
| CREBBP | CREB binding protein | 11.6 | 21.8 | −0.9 | TF |

DOI: https://doi.org/10.7554/eLife.34817.009

activities) fall into this class. These are shown in *Table 1*. To validate results from the RNA-sequencing, quantitative RT-PCR was conducted on five of the genes listed in *Table 1*. These were *CITED*, *IRF9*, *SNAI1*, *EP300* and *HDAC9*. In all cases the trends for fold-changes were the same as in RNA-seq and in most cases they were very close in actual values (*Figure 6*).

It is interesting to note from *Table 1* that only four of the TFs and all of the CMs exhibit downregulation after ATOH1 transduction. The vast majority of the TFs are upregulated. Among the upregulated TF genes, six are of particular note (ATOH1, POU4F3, RAX, ZNF837, GTF2IRD2 AND MEF2B) since they are induced from close to zero to a significant level of transcript abundance (abundance levels are shown in *Table 1*). The remaining TFs appear to already be present at significant abundance levels in gentamicin-treated controls.

19 putative HC markers (ACTC1, ATOH1, CCDC60, CES2, ENO2, EPS8L2, HIST3H2A, JAG2, OBSCN, ODF3B, PKN3, POU4F3, RABL2A, RAX, SYTL1, SYT7, TAS1R1, UBXN11, UNC5A) and six putative downstream targets of ATOH1 (EPS8L2, JAG2, OBSCN, PKN3, POU4F3, UNC5A) (*Cai et al., 2015*), plus 4 markers of HC bundles (EPS8L2, GPI, PFKL, UBA7) (*Shin et al., 2013*) show significant upregulation in the ATOH1-treated samples. *MYO7A*, is also overexpressed in *ATOH1*-transduced tissues compared to all three controls but did not reach statistical significance. These markers of HC maturation are listed in *Supplementary file 1*. A large number of additional established hair cell markers (>150) exhibit consistent upregulation in all three comparative analyses, but failed to pass the statistical threshold of significance (see *Supplementary file 1*). Overall, of the 1375 putative HC markers we searched for within the upregulated gene expression dataset, a total of 335 were detectable and upregulated. These data support the contention that ATOH1 upregulation is driving towards an incomplete program of HC differentiation.

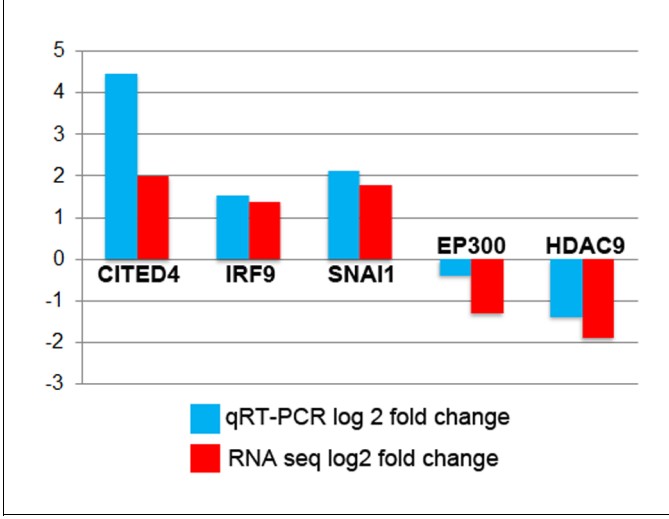

**Figure 6.** Average RQ values in Log2 fold-change for ATOH1 treated versus controls from Taqman (blue) compared to Log2 fold-change measured by RNA-seq (red).
DOI: https://doi.org/10.7554/eLife.34817.008

Enrichment analysis using ToppGene Suite (*Chen et al., 2009*) showed that GO biological processes such as muscle filament sliding and muscle system process were significantly enriched (FDR < 0.01). These processes included actin, myosin and troponin genes which suggests that hair cell bundles might be forming in the ATOH1-transduced tissues. However, most of these genes encoded relatively low abundance transcripts within the total dataset. Interestingly, there was also a significant enrichment of genes with TF binding sites for TCF3 and MEF2A/B transcription factors (FDR < 0.01). TCF3 is a known ATOH1 co-factor (*Masuda et al., 2012*) and is overexpressed in the ATOH1-transduced tissues compared to all three controls but failed to reach statistical significance. MEF2A is not differentially expressed within our data but, as noted above, MEF2B is induced three-fold from a basal level in ATOH1-transduced tissues. These may represent possible potentiators of ATOH1 function.

This unbiased transcriptomic analysis further supports the premise that manipulation of cell fate with gene modification enhances the conversion of supporting cells to a transcriptional signature that overlaps with known markers of the hair cell phenotype.

## Discussion

The objective of the study was to determine whether supporting cells in the vestibular sensory from adult humans can be induced to generate cells with hair-cell like features. As we illustrated previously (*Taylor et al., 2015*) the number of hair cells normally present in the vestibular sensory epithelia of humans may vary widely mainly due to ageing (and genetic variations that likely affect how susceptible to the effects of ageing different individuals might be). Consequently, the initial goal was to try to ablate as many hair cells as possible to create conditions that were essentially the same for all samples, with almost no hair cells present, and generate an epithelium composed predominantly of supporting cells. This was the starting point to test the capacities of supporting cells and also to remove possible confounds presented by any remaining hair cells. This condition also would mimic something similar to what we (*Taylor et al., 2015*) and others (*Wright, 1983*) have shown to be the situation in the vestibular sensory epithelia of elderly people who might be the ones to benefit from a regenerative strategy were one available. The prolonged (48 hr), high dose (2 mM) gentamicin treatment we used was designed to achieve this condition and the results indicate that this was largely accomplished.

Previous work has suggested that supporting cells in the mature utricular maculae of mice can be induced to convert towards a hair cell fate (*Lin et al., 2011*; *Staecker et al., 2007*). The present results demonstrate, not only from expression of hair cell marker proteins but also from analysis of expressed genes, that a similar plasticity of the supporting cell phenotype is retained in the vestibular tissues of humans. In our work, we made no distinction between samples on the basis of age but it has been suggested that age may influence the ability of supporting cells to regenerate lost hair cells. In mammals, a capacity of post-mitotic supporting cells in immature early postnatal (mouse) tissue to respond to regenerative signals or to generate new hair cells is dramatically reduced in the mature tissue of pre-weaner juveniles or young adults (*Burns and Stone, 2017*; *Gu et al., 2007*). The number of cells in the mature vestibular system that have a 'stem cell'-like capacity to form spheroids also declines with age (*Oshima et al., 2007*). However, there is nothing to suggest that in the mature inner ear the activities of supporting cells are affected by age. In the present case, all our material was taken from adults the majority of whom were at the older end of the age range (mean 50.4 years, median 51). Yet in all cases where it was examined, supporting cells retained their capacities to effect repair of the epithelium and maintain tissue integrity. Loss of hair cells was accompanied by concurrent closure of the lesion by expansion of supporting cells, initially with the formation of characteristic scar formations (*Meiteles and Raphael, 1994*; *Li et al., 1995*; *Taylor et al., 2015*). The supporting cells also removed apoptotic bodies efficiently and acted to extrude damaged hair cells, in a manner very reminiscent of the similar processes occurring in vivo in animals following lethal damage to vestibular hair cells. In our earlier work we also showed through dye loading experiments the continuing viability of the supporting cell syncytium in tissue taken from the same population of patients as in this study, and in which there was significant hair cell loss. (*Taylor et al., 2015*). The retention of these capacities and activities seemingly regardless of age or ex vivo maintenance would point to the health of the tissue used in this work and also to the likelihood that the capacity of supporting cells to be converted is maintained throughout life in people. This is

consistent with our earlier observations (*Taylor et al., 2015*) of occasional cells bearing immature hair bundles in the utricular maculae of elderly patients where most mature hair cells had been lost 'naturally', suggesting a very low level of spontaneous hair cell generation may be ongoing in the human vestibular sensory epithelia. Interestingly, it has been suggested that in birds, the capacity to regenerate hair cells from the supporting cell population continues unabated throughout life (*Krumm et al., 2017*).

We found that viral vector-mediated transduction with ATOH1 resulted in generation of highly significant numbers of myosin VIIA positive cells but the use of TAPI1 alone was not as successful. Since the ages of the donors of the tissues used in the TAPI1 experiments was not significantly different from those that were transduced with ATOH1 age is not a factor underlying the difference in the number of cells expressing the hair cell marker protein generated by the treatment regimes. TAPI1 has been shown to be an inhibitor of the Notch pathway that can induce hair cell regeneration in adult mouse utricles (*Lin et al., 2011*) and we used TAPI1 at a concentration the same as that used in that study. Other, γ-secretase inhibitors (GSIs) DAPT (*Lin et al., 2011*) and LY411575 (*Mizutari et al., 2013*) have also been used to regenerate hair cells in murine inner ear tissues. We performed a small number of studies with both of these GSIs, again using doses reported in the literature to be effective in mouse tissue, but neither produced significant numbers of cells expressing myosin VIIa. However, the number of samples examined was insufficient to draw any definitive conclusion as to their efficacy in inducing generation of hair cells. It may also be that higher concentrations of agents that inhibit the Notch pathway are required in the human tissue than in mouse and this may warrant further investigation.

Cells expressing myosin VIIA+ did not form organised hair bundles. The emergence of numerous cells bearing bushy bundles of microvilli in tissues in which Myo VIIa positive cells were generated but not in tissue maintained in culture for similar period but in which only hair cell ablation had occurred, further confirms phenotypic changes to supporting cells. These apical surface characteristics bear a striking resemblance to 'hair-cell-like' cells generated from human embryonic stem cells as described by *Ronaghi et al. (2014)* as well as those defined as hair cells generated in the mouse organ of Corti by genetic co-manipulation of Atoh1 and P27$^{Kip1}$ (*Walters et al., 2017*). However, it would appear that additional factors or further manipulations are required to generate fully differentiated hair cells, a contention supported by our findings that several genes that have been reported to be part of the cascade that follows *Atoh1* expression during the initial embryonic formation of hair cells are not expressed. A similar conclusion is drawn from a recent report (*Walters et al., 2017*) that *Atoh1*-based hair cell regeneration therapies in mouse cochleae may be enhanced by genetic manipulation of p27$^{kip1}$, Gata3 or Pou4f3. From our transcriptional analysis we have identified transcription factors and chromatin modifiers that may play roles in the ATOH1 induced cascade. However, the most glaring absentee from this cascade is GFI1 which is believed to lie downstream of ATOH1 and POU4F3 (*Hertzano et al., 2004*) in the (still ill-defined) TF network that specifies a mature hair cell. In mice lacking *Gfi1* 'hair cells' are generated during development but they show morphological abnormalities and fail to form hair bundles (*Wallis et al., 2003*). Also, in embryoid bodies derived from mouse stem cells the co-expression of *Gfi1* and *Pou4f3* as well as *Atoh1* was required to generate cells with hair bundle-like structures (*Costa et al., 2015*). Since there was significant up-regulation of *POU4F3* expression following the *ATOH1* transduction of the human utricular maculae, it would point to *GFI1* expression as an interesting candidate gene necessary for inducing hair cell differentiation. Gfi1 has been reported to act as a transcriptional regulator acting through chromatin modification (*Duan et al., 2005*). Identification of genes regulated by Gfi1 would also be of interest in a search for factors to induce regeneration of functional hair cells.

The expression of GFP showed that the supporting cells in the human utricular macula can be transduced with quite high efficiency with the Ad2 viral vector, consistent with prior reports (*Kesser et al., 2007*). Efficient transduction of the supporting cells in the human utricle can also be accomplished with a synthetic AAV vector Anc80L65 (*Landegger et al., 2017*). This indicates that other genetic manipulations of human vestibular sensory epithelia are possible and could potentially form the basis for other therapeutic interventions. One known receptor for adenovirus is αvβ5 integrin (*Lyle and McCormick, 2010*; *Wickham et al., 1993*) and in other work (Taylor, Hussain and Forge, in preparation) using RT-PCR screening we have found both these integrin subunits to be expressed in the human vestibular epithelia although as yet we do not know if they are partnered. Nevertheless, this may be the basis of the efficient transduction with adenoviral vectors. However,

some cells that expressed GFP did not appear to express ATOH1 and some cells that expressed ATOH1 did not show expression of GFP. One reason for this maybe that the two genes were driven by independent CMV promoters so may not have always been expressed together or they were expressed at differing levels and sometimes below the level of detection.

The generated cells expressing Myo VIIa, as well as the cells bearing bushy microvillar bundles, were often closely clustered together, sometimes seemingly in contact with each other. The pattern of distribution of hair and supporting cells that is normally present in the utricular macular where each hair cells is surrounded and separated from it neighbours by intervening supporting cells was not restored. This may imply that the signalling pathways through which differentiating hair cells instruct their immediate neighbours to adopt a different, supporting cell, fate from themselves may not be properly active in these conditions where hair cells are induced.

## Conclusions

Here we demonstrate the plasticity of the human vestibular epithelia via manipulation of developmental pathways using a viral vector to transduce supporting cells. The capacity of this vector to incorporate ATOH1 into sufficient supporting cells and subsequently yield large numbers of myosin VIIa+ cells supports our contention that it is possible to regenerate damaged epithelium and offers a therapeutic intervention to balance disorders caused by hair cell loss.

# Materials and methods

**Key resources table**

| Reagent type (species) or resource | Designation | Source or reference | Identifiers | Additional information |
| --- | --- | --- | --- | --- |
| Biological sample | Ad2 -CMV ATOH1:CMV-GFP viral vector | Kesser et al., Laryngoscope 118:821–831 | | |
| Antibody | anti-myosin VIIa | Developmental Studies Hybridoma Bank | DSHB myo7a138-1 | (1:100) |
| Antibody | anti-espin | J. Bartles (Gift) | | (1:50) |
| Antibody | anti sox 2 | Abcam | ab97959 | (1:100) |
| Antibody | anti-ATOH1 | Aviva Bio Systems | ARP 32365_P050 | (1:100) |
| Antibody | FITC, TRITC secondaries | Sigma | | (1:200) |
| Other | Vectorshield with DAPI | Vector Laboratories | H-1200 | |

## Collection and maintenance of tissue

Human vestibular tissue was obtained as previously described (*Taylor et al., 2015*). Briefly, utricular maculae (utricles) and sometimes cristae were collected anonymously following informed consent from patients undergoing excision of vestibular schwannoma (acoustic neuromas) via a trans-labyrinthine approach. Excised tissue was transported from participating hospitals in medium used for long term culture.

Each explant culture was incubated at 37°C in a 5% $CO_2$ atmosphere in one well of a 24 well plate, free floating in 0.5 ml Minimal Essential Medium (MEM) with glutamax (Gibco), 1% HEPES (N-2-hydroxyethylpiperazine-N-2ethanesulfonic acid) and 10% fetal bovine serum (Hyclone). The medium was supplemented with ciprofloxacin and amphotericin B to prevent contamination. To ablate hair cells, tissue was exposed to 2 mmol/L gentamicin for 48 hr and subsequently rinsed thoroughly with fresh medium before maintenance for up to 21 days with 50% of the medium changed on alternate days.

## Ad2-ATOH1-GFP transduction of cells

Replication deficient second generation adenovirus (Ad2) with deleted E1, E3, pol, pTP regions was used as vector (*Kesser et al., 2007*; *Kesser et al., 2008*). The Ad2 virus contained two expression cassettes each driven by the human cytomegalovirus promoter: *CMV-ATOH1* and *CMV-GFP*. Aliquots of Ad2-ATOH1-GFP viral vector stock were stored at −80°C until required. Following gentamicin treatment, tissue was rinsed three times in serum-free medium containing ciprofloxacin and

incubated for up to 24 hr but no less than 18 hr in 200 µL of serum-free medium with a dilution of Ad2-ATOH1-GFP to give $1 \times 10^8$ total particles per ml. Tissue was then rinsed five times with MEM with glutamax with serum to halt the transduction and maintained in this medium for a further 17 days. Cultures were rinsed and processed for immunohistochemistry or electron microscopy as described below.

The Notch pathway was inhibited using the TACE inhibitor TAPI-1. Gentamicin-treated samples were maintained in medium +50 µM TAPI-1 (Sigma-Aldrich, Poole) throughout the duration of culture. Initially TAPI-1 was dissolved in DMSO and further diluted with medium to give a stock solution of 1 mM that was stored at −20°C. These utricles were rinsed, fixed and processed for immunohistochemistry or electron microscopy.

Six lesioned utricles were transduced with Ad2-ATOH1-GFP and then incubated continuously in medium containing TAPI-1 as for the above samples. They were processed for immunohistochemistry or electron microscopy.

## Tissue processing

Tissue for immunohistochemistry was fixed in 4% paraformaldehyde (PFA) in phosphate buffered saline solution (PBS) for 90 min. The majority of samples were prepared as whole mounts with a small number prepared for cryosectioning. Whole mounts were rinsed in PBS and permeabilized using 0.5% Triton X-100 for 20 min and placed in blocking solution (10% goat serum, 0.15% Triton in PBS). Tissue was incubated overnight at 4°C in primary antibody, rinsed thoroughly in PBS and then incubated for 2 hr at room temperature with the appropriate secondary antibody conjugated to a fluorophore. Primary antibodies used were: mouse monoclonal against myosin VIIA (Developmental Studies Hybridoma Bank; myo7a 138–1) used at a dilution of 1:100; a rabbit polyclonal against espin (a kind gift from J Bartles) used at 1:50; a rabbit polyclonal against sox 2 (Abcam, ab 97959) at 1:100 and a polyclonal against Atoh1 (Aviva Systems Biology, ARP 32365_P050) at 1:100. A Tyramide signal amplification kit (Molecular Probes) was used according to manufacturer's protocol to amplify the Atoh1 labelling in tissue. Tissue was incubated in the appropriate secondary antibody (sheep anti-mouse (Zymed), or goat anti-rabbit (Sigma)) at 1:200. A fluorescent phalloidin conjugate (Sigma) was added at 1 µg/ml to the secondary antibody solution to label filamentous actin. Following staining, utricles were mounted onto slides using Vectashield with DAPI (Vectorlabs) to label nuclei. Samples were examined and images captured with a Zeiss LSM 510 confocal microscope.

For cryosections, fixed tissue was incubated in 30% sucrose solution overnight at 4°C embedded in low-temperature setting agarose and mounted in the required orientation. Cryosections of 15 µm were cut and collected on polylysine coated slides (VWR). Immunolabelling was performed as for whole mounts.

For scanning electron microscopy (SEM) and transmission electron microscopy (TEM) cultured utricles were rinsed and fixed in 2.5% glutaraldehyde in 0.1 mol/L cacodylate buffer for 2 hr and subsequently post-fixed in 1% OsO4 for 1.5 hr. Utricles for SEM, were then processed following the repeated thiocarbohydrazide-osmium procedure (*Davies and Forge, 1987*), dehydrated in an ethanol series and critical point dried. Samples were mounted on support stubs using silver conductive paint and sputter coated with platinum before examination and collection of digital images on a Jeol 6700F instrument. Tissue for thin sectioning was partially dehydrated to 70% ethanol and stained 'en bloc' with uranyl acetate in 70% ethanol before completing dehydration and embedding in plastic. Some immunolabelled whole mount samples were removed from the slides after confocal microscopy, fixed in glutaraldehyde and $OsO_4$ and processed for thin sectioning. Some samples examined by SEM also were prepared for thin sectioning. They were removed from the specimen support stubs into acetone, then into 100% ethanol before embedding in plastic. For all plastic embedded samples, serial thin sections across the entire width of each utricle were collected at a minimum of three depths separated by ca. 50 µm. Sections, some mounted on formvar –coated single slot grids, were stained with aqueous uranyl acetate and lead citrate and examined in a Jeol1200EXII instrument. Digital images were acquired with a Gatan camera. Sections on grids were also examined and imaged in the SEM using back-scatter detection to provide uninterrupted views of the entire width of the section.

## Quantification

Myosin VIIA+ cells were viewed and quantified from z-stacks of confocal images viewed in Image J.. Assessment was made of at least two different fields on a single utricle viewed with a x20 objective, with a random movement in X and Y planes between each field. In each field intact Myo VIIa positive cells with a distinct nucleus were counted in a delineated, measured area of at least 20,000 $\mu m^2$ enclosing continuous intact epithelium as defined by phalloidin labelling of cell-cell junctions at the luminal surface, and excluding regions where the epithelium was folded over on itself, or was significantly disrupted, which occurred in several samples during prolonged incubation and processing due to the friability of the tissue and detachment of the epithelium from the underlying mesenchyme. At each location, each individual optical section through the entire depth of the epithelium was analysed. Cell counts were normalised to a unit area of 10000 $um^2$.

## Data analysis

Using Prism4 GraphPad, discrete comparisons were made between each individual treatment group and the gentamicin only treated cultures maintained for similar lengths of time (control) using t-tests. ANOVA with Tukey correction was used to assess whether there were significant differences between the treatment groups.

## Sample size calculations for transcriptome analysis

Previous mRNA-Seq data derived by the Lovett group (*Ku et al., 2014*) were used to calculate the median coefficient of variation in gene expression across samples when FPKM (Fragments Per Kilobase Of Exon Per Million Fragments Mapped) values > 1 were used. Based on the model published by *Hart et al. (2013)*, two biological replicates per treatment group are needed to detect a 2-fold difference with 80% power and 0.1 type 1 error, if 6 samples are multiplexed per lane (~1000 read counts per transcript). Based on these calculations, in this study we sequenced 2 replicates for the treatment group and 2 replicates for the control group (14–18 days post gentamycin only). Two additional control samples (2dpg and 8 dpg) were available and therefore these were also used in the statistical analysis as described in the methods section. Three untreated samples were also sequenced and used in the ANOVA model but no comparisons with these were performed for the purposes of this study.

## Transcriptome analysis

Six whole utricle samples (as pure as possible) were used for transcriptome analysis: Ad2-ATOH1-GFP transduced utricles cultured for 5 days after gentamicin treatment (two biological replicates), a utricle cultured for 2 days post gentamicin (early time-point, first control), one utricle cultured for 8 days post gentamicin (mid time-point, second control) and two utricles cultured for 14–18 days post gentamicin (late time-point, these two samples were grouped and used as the third control group). Following treatment, all utricles were collected into RNA-later. RNA was extracted using Zymo Research Quick-RNA MicroPrep kit. Libraries were prepared using Illumina Truseq Stranded mRNA Library Prep kit. 75 bp paired-end sequencing was performed on the Illumina HiSeq platform by service provider CNAG, Barcelona, Spain. Raw reads in fastq format were trimmed to remove adapter sequences and low-quality bases (Q < 30) from the 3' end of reads using Cutadapt v1.9 (*Martin, 2011*), then aligned against hg38 using Tophat v2.1 (*Kim et al., 2013*) and expression values (FPKMs) were generated using Cufflinks v2.1 (*Trapnell et al., 2010*) based on ensembl87 annotations. A filtering step was applied with at least one sample to be ≥0.5 FPKM. All sequencing data from all of these samples have been deposited in NCBI GEO (number applied for).

The utricle samples were not age-matched and due to possible genetic variability, Ad2-ATOH1-GFP transduced samples were compared against different control samples treated with gentamicin only. Three comparisons between Atoh1-transduced utricle samples against gentamicin-treated only samples were performed using one-way Anova on Partek Genomics Suite software: a) Atoh1-transduced versus 2 days post-gentamicin control sample; b) Atoh1-transduced versus 8 days post-gentamicin control sample and c) Atoh1-transduced versus 14–18 days post-gentamicin group of controls). Biological replicates and the group of control gentamicin-treated samples for 14–18 days had an average spearman's rho of 0.92. Statistical significance was considered when p<0.05 and $-2 \leq$ fold change≥2 and fold changes were in the same direction across the three comparisons.

Statistically significant differentially expressed genes were uploaded in ToppGene (*Chen et al., 2009*) to identify enriched GO biological processes (FDR < 0.05) and GeneMANIA (*Warde-Farley et al., 2010*) to identify literature-supported interactions.

## Quantitative RT-PCR

Taqman assays for *CITED, IRF9, SNAI1, EP300* and *HDAC9* (Hs 00388363, Hs00196051, Hs00195591, Hs00914223 and Hs-1081558) were purchased from ABI and were run in technical triplicates on an ABI QuantStudio Real Time PCR System under manufacturer's standard parameters for comparative $C_T$ analysis. Total polyA+ RNA from two *ATOH1* utricle transduction experiments (5 days post gentamicin) were separately converted into cDNA. These were then pooled and constituted the *ATOH* samples. Total polyA+ RNA from three control utricle samples (2 days, 8 days and 14 days post gentamicin) were separately converted into cDNA. These were then pooled and constituted the control samples. Amplifications were normalized to a GAPDH internal control (ABI Taqman assay Hs99999905).

## Acknowledgements

The work was supported by project grants from the UK Medical Research Council, the Dunhill Medical Trust and the Rosetrees Trust. The human vestibular tissue was harvested by Shakeel Saeed (Royal National Throat Nose and Ear Hospital, UCLH), Patrick Axon, Neil Donnelly and James Tysome (Addenbrooke's Hospital, Cambridge), Richard Irving, Peter Monksfield and Chris Coulson (Queen Elizabeth Hospital, Birmingham) and Simon R Freeman and Simon K Lloyd (Manchester Royal Infirmary and Salford Royal Infirmary)

## Additional information

### Funding

| Funder | Grant reference number | Author |
| --- | --- | --- |
| Medical Research Council | G1000068 | Ruth Rebecca Taylor<br>Andrew Forge |
| Dunhill Medical Trust | R395/1114 | Andrew Forge |
| Rosetrees Trust | M58-F1 | Andrew Forge |

The funders had no role in study design, data collection and interpretation, or the decision to submit the work for publication.

### Author contributions

Ruth Rebecca Taylor, Conceptualization, Data curation, Formal analysis, Investigation, Methodology, Writing—original draft, Project administration, Writing—review and editing; Anastasia Filia, Formal analysis, Investigation, Methodology, Writing—review and editing; Ursula Paredes, Investigation; Yukako Asai, Resources, Investigation; Jeffrey R Holt, Resources, Investigation, Methodology, Writing—review and editing; Michael Lovett, Data curation, Formal analysis, Investigation, Methodology, Writing—review and editing; Andrew Forge, Conceptualization, Formal analysis, Funding acquisition, Investigation, Methodology, Writing—original draft, Project administration, Writing—review and editing

### Author ORCIDs

Ruth Rebecca Taylor https://orcid.org/0000-0001-7359-1604
Andrew Forge http://orcid.org/0000-0002-0995-0219

### Ethics

Human subjects: Ethical approval from NHS Health Research Authority, NRES Committee London–Surrey Borders (REC reference 11/LO/0475; IRAS project ID 73422). Tissue was collected

anonymously with informed consent of the patient for tissue harvesting and publication of the results of the study.

## Decision letter and Author response
Decision letter https://doi.org/10.7554/eLife.34817.015
Author response https://doi.org/10.7554/eLife.34817.016

## Additional files

### Supplementary files
• Supplementary file 1. Known and putative hair cell markers that exhibit upregulation in gene expression upon Atoh1 transduction into human sensory epithelia. The left column lists gene names followed by gene descriptions. AB T is the average transcript abundance in the Atoh1 treated samples (as FPKMs). AB C is the average abundance in the control samples. LOG2 is the log base2 fold change of Atoh1 transfected compared to the controls. REFERENCE indicates the published results showing supportive evidence that the given gene is a putative hair cell marker (1) *Cai et al. (2015)*; (2) *Scheffer et al., 2015*; (3) *Shin et al. (2013)*. Names in bold are genes that show a > 2 fold change in expression (and p<0.05) across all three comparisons (they are part of the 441 differentially expressed genes, see materials and methods and results). Genes with an asterisk (*) are putative Atoh1 downstream targets based on Groves et al publication. Genes in plain text and above the thick bottom line show statistically significant upregulation in all comparisons, but only pass > 2 fold in at least one comparison. Genes below the thick bottom line do not pass any of the above significance thresholds and show at least upregulation in all comparisons.
DOI: https://doi.org/10.7554/eLife.34817.010
• Transparent reporting form
DOI: https://doi.org/10.7554/eLife.34817.011

### Data availability
All sequencing data from all of these samples have been deposited in NCBI GEO (accession number: GSE109320).

The following dataset was generated:

| Author(s) | Year | Dataset title | Dataset URL | Database, license, and accessibility information |
|---|---|---|---|---|
| Forge A, Taylor RR, Filia A, Paredes U, Asai Y, Holt JR, Lovett M | 2018 | Regenerating hair cells in human vestibular sensory epithelia | www.ncbi.nlm.nih.gov/geo/query/acc.cgi?acc=GSE109320 | Publicly available at the NCBI Gene Expression Omnibus (accession no. GSE109320) |

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
