## [Decision Letter]

Thank you for submitting your article "Regenerating hair cells in human vestibular sensory epithelia" for consideration by *eLife*. Your article has been reviewed by Andrew King as the Senior Editor, Tanya Whitfield as the Reviewing Editor and three reviewers. The following individual involved in review of your submission has agreed to reveal his identity: Jordi Llorens (Reviewer #1).

The reviewers have discussed the reviews with one another and the Reviewing Editor has drafted this decision to help you prepare a revised submission.

Summary:

This article evaluates the potential for inducing hair cell regeneration in cultured utricles from adult humans. It is concluded that *ATOH1* transduction stimulates an incipient regeneration of hair cells in utricles damaged in vitro by gentamicin, but that the stimulus is insufficient to generate maturely differentiated hair cells. Similar, but quantitatively smaller, effects were obtained by pharmacological blockade of the Notch signalling pathway. The study of the gene expression changes after *ATOH1* treatment open a powerful way to future progress in understanding the requirements for hair cell regeneration.

Essential revisions:

As you will see, all three reviewers are largely positive about the study, but each has a number of queries regarding the data and methodology, particularly relating to the cell counting methods and statistical analysis, together with age and disease status of the patients. Additional validation and retrospective analysis of existing data are suggested where this would clarify the interpretation and conclusions. The full reviews are appended below.

Reviewer #1:

Thanks to previously described consortium of surgeons and protocol (Taylor et al., 2015), these cultures represent a unique opportunity to study candidate therapeutic approaches in the adult human tissues where these treatments would be used at the end. This major strength bears inevitably the major weakness of the study, the difficulty in obtaining homogeneous and large groups of samples for complete analysis. The model imposes an experimental design of one sample at a time, so experimental groups are constructed over months. Under these circumstances, one must admit that some questions that would require an answer in animal studies cannot be realistically asked in this model. However, there are several important aspects that should be somehow addressed or at least fairly discussed by the authors.

1) The first question is on the viability of the hair cells in vitro, and the basal conditions on which the lesion/regeneration model is compared with. In subsection “Untreated utricles” one reads "Hair cells survived in vitro in untreated samples maintained in explant culture for up to 21 days", but also "Condensed, misshapen remnants of dying cells also immunolabelled positively for myosin VIIa". The second sentence reveals that the culture conditions are suboptimal and progressively cause hair cell death, thus denying the first sentence. Therefore, to allow for a good appraisal of the damage and regeneration capacities of the utricles in the model, the article should include data of hair cell density in the cultures before gentamicin treatment, and in untreated utricles after 18-20 days in culture. A fair estimation of these densities should be possible retrospectively from stored images of the experiments reported, or from ongoing experiments. Otherwise, this aspect should be discussed.

2) The presence of elongated microvilli (Figure 5A) in damaged utricles treated with *ATOH1* is shown. However, their absence (or smaller presence) in damaged untreated utricles or in non-damaged utricles is not shown. The authors must have this evidence to support their conclusion.

3) The *MYO7A*+ data are analyzed by multiple t-test of damaged+treated groups versus a common single damaged-untreated group. The data should instead be analyzed by one-way ANOVA followed by an appropriate post-hoc test for multiple comparisons.

4) The age of the donors in each group must be shown and compared, to eliminate the possibility that the observed differences among groups are due to age differences. In fact, the image in Figure 1A shows a density of hair cells much lower than that expected in young adult mammals, suggesting that this utricle suffers hair cell loss, probably related to the age of the donor (Taylor et al., 2015). May this explain the unexpected finding that ATOH1+TAPI-1 utricles showed less *MYO7A*+ cells than the ATOH1 alone utricles?

Reviewer #2:

The submitted manuscript by Forge's research group characterizes the response of *Atoh1* gene transfer into the cultured human utricles. This is an important topic because there are very few studies describing how the human inner ear regenerates in response to manipulation. Overall, the study is large descriptive and insufficient or inconsistent information was given on both the *Atoh1* transfection and Notch inhibition experiments, making it difficult for the readers to interpret the results and make meaning conclusion.

First, it is unclear how variably healthy the tissues are when first harvested, as no hair cell counts, or low/mag high magnification pictures were given. A previous study from this group (Taylor et al., 2015) has characterized missing hair cells and occasion hair cells with damaged stereocilia, and such variable composition (hair cell number, supporting cell number, and hair cells missing stereociliary bundle) can likely affect the final results of the experiments. For instance, hair cells without bundles may survive aminoglycoside treatment and can be directly transfected and re-grow stereocilia (Yang et al., 2012). Also, given the substantial size of the human utricle, hair cell counts from 2000 microns also seem inadequate. It would be important for the authors to reason the quantification method and describe how those areas chosen can reasonably represent the whole sensory epithelium. Also, are striolar vs extrastriolar regions separately analyzed?

Secondly, the efficiency of transfection will need to be more thoroughly described. At present, only high mag pictures showing some transfected supporting cells are shown in Figure 3. Are there any hair cells transfected early on to suggest that surviving hair cells were transfected? What is the extent of *Atoh1* expression and degree of colocalization with GFP+ cells? Right now, only 2 GFP cells are shown in the utricle and one expresses *Atoh1*. Figure 3D describes the crista, but epithelium appears flat isn't shaped convex as in the crista. Again, more thorough quantification with low mag picture to go with the high mag pictures would be essential. Similarly, it is unclear how TAPI-treated tissues are quantitatively analyzed, as no pictures were shown, and only small areas were reportedly analyzed. How was the dose of TAPI chosen? How does the author decide that is the optimal dose for human utricles? The authors also discussed the use of DAPT (they said DAPI) and LY411575 and found no definitely effects. Thus, it is difficult to determine if Notch inhibition (or when combined with *Atoh1*) is less effective in inducing new hair cells or not.

With regards to SEM and TEM analyses in Figure 5, it is unclear how the authors determine that the pictures represent the GFP-transfected cells without co-labeling. As stated earlier, the information on the rate and cell types transfected and also health of harvested tissues will help the readers determine the meaning of these TEM and SEM analyses.

Lastly, the RNAseq experiments will benefit from further validation of genes up- and downregulated.

Reviewer #3:

This is an important paper that provides evidence for regeneration of hair cells in the vestibular epithelium in aged human tissue. This reviewer appreciates the rarity of being able to analyse such tissue and the authors have done an excellent job in getting the most out of this valuable tissue.

Discussion section: "the generated cells are sometimes seeming in contact with each other" Do the authors have any transmission electron microscopy data? If so, it would be good to include it, but of course, it is not necessary to do new experiments. Are the authors suggesting here that the newly generated cells are derived from a single clone?

Subsection “Quantification”: For the cell counts, can the authors identify which region of the utricular macula the counts were taken? It would make a difference whether the analysis was done in the striolar (central) region versus the extrastriolar (peripheral) region, as the striolar region is damaged first and develops first during normal development. I think this would be an important thing to analyse if the confocal stacks can still be visualised.

Figure 6B: This reviewer is not certain that the apoptotic body is inside a supporting cell, not unless the supporting cell is also dying. The cytoplasm of the cell in question, although it appears to be touching the basement membrane, is much lighter than the surrounding cells, which are indeed supporting cells, as one can tell by the presence of secretory granules within their cytoplasm and their more clumped heterochromatin. Although it seems unlikely that one hair cell would endocytose another, stranger things have happened during cell death.

Figure 6F: Are the rod-like inclusions the same as what have been termed "cytocauds"? Can the authors speculate upon whether these are re-absorbed hair bundles?

[Editors' note: further revisions were requested prior to acceptance, as described below.]

Thank you for resubmitting your work entitled "Regenerating hair cells in human vestibular sensory epithelia" for further consideration at *eLife*. Your revised article has been evaluated by Andrew King (Senior Editor), Tanya Whitfield (Reviewing Editor), and one reviewer.

We felt that you had addressed comments from reviewers 1 and 3 satisfactorily, but reviewer 2 had asked for further validation of your RNAseq data. In your response, you stated that this had been 'clarified'. However, this did not actually seem to have been addressed anywhere. Suitable experiments would include qPCR and/or in situ hybridisation or immunohistochemistry for a selection of the genes that you have identified via RNASeq as up- or down-regulated after *ATOH1* mis-expression. This additional validation was listed as an essential revision in the initial decision letter, and this requirement still stands. Please see the detailed comments from the reviewer below.

Reviewer #2:

The authors have done a decent job addressing the reviewers' concerns to improve the candidacy of this manuscript. Specifically, the methods used to quantify hair cells and the variability of hair cell number are described and the use of a supratherapeutic dose of aminoglycoside to ablate "all" hair cells explained. However, it is still somewhat perplexing that authors think that hair cells without bundles can take up aminoglycosides and be ablated.

Nonetheless, more convincing pictures of ATOH1-GFP, Myosin7+ hair cells were shown, indicating that supporting cells are competent. However, Figure 3D still shows that both actin and DAPI in blue (I think this is a typo). Lastly, it is stated that RNAseq experiment results should be validated, this was not addressed even though the author states "clarified". Since mechanism of *Atoh1* can be viewed as a strength of this paper, validation of these data will help its candidacy.

---

## [Author Response]

Reviewer #1:

*[…] 1) The first question is on the viability of the hair cells* in vitro*, and the basal conditions on which the lesion/regeneration model is compared with. In subsection “Untreated utricles” one reads "Hair cells survived* in vitro *in untreated samples maintained in explant culture for up to 21 days", but also "Condensed, misshapen remnants of dying cells also immunolabelled positively for myosin VIIa". The second sentence reveals that the culture conditions are suboptimal and progressively cause hair cell death, thus denying the first sentence.*

The point we are making is that it is possible to maintain the tissue in good condition for up to ca. 28 days at which point sometimes there are indications in some cultures of incipient degeneration. This provides the time scale for the incubation periods to be used subsequently (21-22 days) to explore regenerative potential.

Amended to say:

“Hair cells survived in vitro in most untreated samples maintained in explant culture for 28 days although in some cultures maintained for this time, condensed, misshapen remnants also labelled positively for myosin VIIa suggesting a possible incipient deterioration in the cultures by this time (Figure 1D). This defined a period of 21-22 days for an optimal total time of incubation in subsequent experiments, a period of sufficient length to cover that over which spontaneous regeneration of hair cells occurs in the vestibular organs in vivo in guinea pigs (Forge et al. 1993,1998), chinchillas (Lopez et al. 1997) and mice (Kawamoto et al., 2009).”

Therefore, to allow for a good appraisal of the damage and regeneration capacities of the utricles in the model, the article should include data of hair cell density in the cultures before gentamicin treatment, and in untreated utricles after 18-20 days in culture. A fair estimation of these densities should be possible retrospectively from stored images of the experiments reported, or from ongoing experiments. Otherwise, this aspect should be discussed.

There was no intention to make comparisons directly between the tissue before treatment to ablate hair cells and that after the application of a potential regenerative procedure, nor between that condition and the number of hair cells that would have been present if no ablation had occurred. We have tried to clarify this in what we have added in the Discussion section, which hopefully has removed any misunderstanding as to the comparisons being made.

We now add in the opening paragraph of the Discussion section:

“The objective of the study was to determine whether supporting cells can be induced to generate cells with hair-cell like features [we have also included a phrase in the last paragraph of the Introduction to emphasise this]. As we illustrated in our previous paper the number of hair cells in the vestibular sensory epithelia of humans may vary widely mainly due to ageing (and genetic variations that likely affect how susceptible to the effects of ageing different individuals might be). Consequently, the initial goal was to try to ablate as many hair cells as possible to create conditions that were essentially the same for all samples, with almost no hair cells present to create an epithelium composed predominantly of supporting cells. This was the starting point to test the capacities of supporting cells and also to remove possible confounds presented by any remaining hair cells. This condition also would mimic something similar to what we and others (Wright, 1983) have shown to be the situation in the vestibular sensory epithelia of elderly people who might be the ones to benefit from a regenerative strategy were one available. The prolonged (48h), high dose (2mM) gentamicin treatment we used was designed to achieve this condition and the results indicate that this was largely accomplished”.

2) The presence of elongated microvilli (Figure 5A) in damaged utricles treated with ATOH1 is shown. However, their absence (or smaller presence) in damaged untreated utricles or in non-damaged utricles is not shown. The authors must have this evidence to support their conclusion.

An additional panel has been added to Figure 2 panel I to illustrate the absence of cell surface projections and is referred to in the text:

“The cell surfaces of almost all cells across the epithelium were of similar appearance (Figure 2I), with no surface projections or other structural specialisations, except for dispersed short microvilli, and with a polygonal outline, features characteristic of supporting cells following loss of hair cells.”

3) The MYO7A+ data are analyzed by multiple t-test of damaged+treated groups versus a common single damaged-untreated group. The data should instead be analyzed by one-way ANOVA followed by an appropriate post-hoc test for multiple comparisons.

We are aware that ANOVA is the appropriate test for multiple pairwise comparisons. We did not use the t-tests for this purpose and we did not comment upon significance levels between the three different treatment groups. Our initial aim was to determine whether any of the individual treatment conditions produced a significant difference from the control (hair cells ablated, no subsequent treatment, tissue maintained for a prolonged period post hair cell ablation). Each treatment condition is independent of the others and therefore the comparisons are discrete and used to discover treatments that induced hair cell generation from supporting cells. We believe independent t-tests are appropriate for such analysis and advice from others with statistical expertise confirmed this. Nevertheless, we did also perform an ANOVA with Tukey correction. This showed additionally that there is no significant difference between the ATOH1 alone and ATOH1+TAPI conditions; i.e. combining treatments does not enhance the ATOH1-induced generation the MyoVIIa +ve cells, a conclusion it was possible to draw from the means presented in the text and the box plots presented in figure 4. ANOVA also showed a significant difference between ATOH1 transduction and TAPI1 treatment. Essentially this also is evident from the box plots. In the text, for clarity and brevity, we did not mention significance levels between treatment groups, but have now included them in the text.

4) The age of the donors in each group must be shown and compared, to eliminate the possibility that the observed differences among groups are due to age differences.

The mean ages and the age ranges for each group are included in the text. There are no significant differences between groups in the ages of the donors of the samples. We have added in the text:

“There was no significant difference in the mean ages of the donors in each treatment group and the age ranges were similar (control: mean 58.1, range 36-81; ATOH1 transduced: mean 55.7, range 41-67; TAPI1: mean 46.8, range 32-64; ATOH1+TAPI: mean 46.7; range 20-71). This indicated that age is unlikely to be a factor underlying the difference between treatment regimes in the in the number of myosin VIIa cells generated”.

However, we would note that we see no reason why age should have an effect on the capacity of supporting cells to respond to signals. We now discuss this in the Discussion section which also addresses the question of the health of the tissue.

“In our work, we made no distinction between samples on the basis of age, but it has been suggested that age may influence the ability of supporting cells to regenerate lost hair cells. In mammals, a capacity of post-mitotic supporting cells in immature early postnatal (mouse) tissue to respond to regenerative signals or to generate new hair cells is dramatically reduced in the mature tissue of pre-weaner juveniles or young adults (Burns and Stone, 2017; Gu et al., 2007). […] Interestingly, it has been suggested that in birds, the capacity to regenerate hair cells from the supporting cell population continues unabated throughout life (Krumm et al., 2017).”

In fact, the image in Figure 1A shows a density of hair cells much lower than that expected in young adult mammals, suggesting that this utricle suffers hair cell loss, probably related to the age of the donor (Taylor et al., 2015). May this explain the unexpected finding that ATOH1+TAPI-1 utricles showed less MYO7A+ cells than the ATOH1 alone utricles?

The means and range of ages for the individuals in each treatment group are included in the text in the Results section:

“There was no significant difference in the mean ages of the donors in each treatment group and the age ranges were similar (control: mean 59.7, range 45-81; ATOH1 transduced: mean 55.3, range 41-66; TAPI1: mean 46.6, range 32-57; ATOH1+TAPI: mean 46.7; range 20-71). This indicated that age is unlikely to be a factor underlying the difference between treatment regimes in the in the number of myosin VIIa cells generated”.

There are no significant differences between the groups in the ages of the tissues donors, but it should be noted as pointed out above that the prolonged (48h), high dose (2mM) gentamicin treatment we used was designed to ablate as many hair cells as possible, to obtain as far as practicable a uniform starting point of a predominantly supporting cell population for all samples to be tested for regenerative potential. In this context, we believe the hair cell number in the tissue when first harvested is not really relevant to the study.

Reviewer #2:The submitted manuscript by Forge's research group characterizes the response of Atoh1 gene transfer into the cultured human utricles. This is an important topic because there are very few studies describing how the human inner ear regenerates in response to manipulation. Overall, the study is large descriptive and insufficient or inconsistent information was given on both the Atoh1 transfection and Notch inhibition experiments, making it difficult for the readers to interpret the results and make meaning conclusion.First, it is unclear how variably healthy the tissues are when first harvested, as no hair cell counts, or low/mag high magnification pictures were given.

We are not clear what is meant by “healthy”, but the suggestion would seem to be that it is based on hair cell numbers. We performed a quality control as best as was possible when harvested tissue arrived in the laboratory to ensure that there was intact epithelium and to assess its quality. About half the samples did not meet a series of criteria and were rejected (We refer to this in the text). As we also state in our earlier paper (Taylor et al., 2015), to which reference is made, some samples, taken at random, were exposed to the dye FM1-43, which is taken up by viable hair cells, to identify whether viable hair cells were present. But also, as we point out in answer to reviewer 1 above, the important criterion is the health of the supporting cells. Hair cell number is no indicator of this; was highly variable between samples; and, as we point out in response to reviewer 1 we tried to and appeared to succeed in ablating almost all hair cells before we initiated examination of potential regenerative regimes. We therefore feel the number of hair cells present at the outset is not really relevant.

But as we again point out in the response to reviewer 1 above and have now included in the Discussion section of the present paper, there are certain criteria by which it is possible to judge that the supporting cells in most of the cultures were healthy even when obtained from elderly donors and/or maintained ex vivo for prolonged periods.

A previous study from this group (Taylor et al., 2015) has characterized missing hair cells and occasion hair cells with damaged stereocilia, and such variable composition (hair cell number, supporting cell number, and hair cells missing stereociliary bundle) can likely affect the final results of the experiments. For instance, hair cells without bundles may survive aminoglycoside treatment and can be directly transfected and re-grow stereocilia (Yang et al., 2012).

We also showed in that earlier paper that FM1-43 was taken up into hair cells in samples which subsequent SEM showed had lost hair bundles from most of the hair cells. FM1-43 is taken up into viable hair cells and thus, such uptake is a measure of hair cell viability. The dye is also used increasingly as a surrogate for aminoglycoside in studies of drug uptake as it enters hair cells via the transduction channel as do aminoglycosides; if FM1-43 enters the cell, then aminoglycoside uptake is likely. Thus, our previous observations would suggest that aminoglycosides can enter hair cells even when the hair bundle is lost. In the present work, as pointed out above in response to reviewer 1, we used a high concentration of gentamicin for a prolonged period to try to ensure that almost all hair cells were ablated – which the results show was achieved.

While the study by Yang et al., and others, have shown that there may be a potential for sub-lethally damaged hair cells to lose hair bundles and re-grow them, in most of those cases where this has been examined in mammals, including the study by Yang et al., the hair cells are retained at the surface of the epithelium and are easily distinguished from supporting cells by their smooth and defined, approximately rounded apical surfaces. In our previous paper we showed this to be the case for hair cells that had lost their hair bundles, were not exposed to gentamicin in culture, and were maintained for up to 4 weeks after harvesting. The absence of these distinctive hair cell surface features in the gentamicin treated tissue in the present work is evident in the newly added Figure 2I. Furthermore, in the study by Yang et al., and other papers where repair has been suggested, the “repaired” hair bundles show characteristic hair bundle morphologies, which, as we point out, we have never seen in the studies described in this paper. In addition, when counting cell bodies labelled for myosin VIIa as a measure of hair cell numbers, care was taken to assess the entire depth of the epithelium so that any hair cell bodies enclosed with the corpus of the epithelium, would be included, but the counts showed that the majority of hair cells were ablated by the gentamicin treatment protocol. These observations suggest few if any of the original hair cells would have been transduced and contributed to the population numbers of myosin VIIa labelled cells following transduction.

Also, given the substantial size of the human utricle, hair cell counts from 2000 microns also seem inadequate. It would be important for the authors to reason the quantification method and describe how those areas chosen can reasonably represent the whole sensory epithelium.

The reviewer draws attention to an error. The hair cell counts quoted and used in analysis are normalised for a unit area of 10,000µm^2^. This has been corrected throughout. We try to clarify the procedure used with amendments to the text in the Materials and methods section.

“Assessment was made of at least two different fields on a single utricle viewed with a x20 objective, with a random movement in X and Y planes between each field. In each field intact Myo VIIa positive cells with a distinct nucleus were counted in a delineated, measured area of at least 20,000µm^2^ enclosing continuous intact epithelium as defined by phalloidin labelling of cell-cell junctions at the luminal surface, and excluding regions where the epithelium was folded over on itself, or was significantly disrupted, which occurred in several samples during prolonged incubation and processing due to the friability of the tissue and detachment of the epithelium from the underlying mesenchyme. At each location, each individual optical section through the entire depth of the epithelium was analysed. Cell counts were normalised to a unit area of 10000µm^2^.”

Also, are striolar vs extrastriolar regions separately analyzed?

We did not distinguish between striolar and extrastriolar regions especially as it was difficult to define these regions after the hair cell loss. As the figures (and hair cell counts) illustrate, most hair cells were ablated and thus it was difficult to locate the striola by a region of hair bundle polarity reversal, and the cells that were generated did not form polarised bundles so there was no line of polarity reversal to define.

Secondly, the efficiency of transfection will need to be more thoroughly described. At present, only high mag pictures showing some transfected supporting cells are shown in Figure 3. Are there any hair cells transfected early on to suggest that surviving hair cells were transfected?

It is likely that if there were surviving, intact hair cells they would be transduced. We have no evidence for this occurrence, but as we show almost all hair cells – cells that labelled for myosin VIIa – surviving at the end of the period of incubation with gentamicin showed features indicating a terminal pathology, most notably the internalised actin bundle that has been suggested to be a structure associated with imminent phagocytosis by supporting cells (Bucks et al., 2017). The number of such cells was small as our reported hair cell counts show and thus, if they survived for the entire period post-transduction are unlikely to have contributed greatly to the outcome.

What is the extent of Atoh1 expression and degree of colocalization with GFP+ cells? Right now, only 2 GFP cells are shown in the utricle and one expresses Atoh1. Figure 3D describes the crista, but epithelium appears flat isn't shaped convex as in the crista. Again, more thorough quantification with low mag picture to go with the high mag pictures would be essential.

The images shown are from frozen sections (as now note in the legend to Figure 2), with which, in our hands, we find immunolabelling for ATOH1 to be more reliable than with whole mounts, possibly because of the need use signal amplification protocols (as described in the Materials and methods section) for detection of the labelling. Unfortunately, this made quantification difficult. However, the transcriptomic analysis shows that at 5 days after transduction there is a highly significant expression level of *ATOH1* as already stated in the text.

The image of the crista has been presented at 90° to the long axis to be consistent with orientation of other utricular maculae for the benefit of readers who are not familiar with the morphology of cristae.

Similarly, it is unclear how TAPI-treated tissues are quantitatively analyzed, as no pictures were shown, and only small areas were reportedly analyzed.

The TAPI 1 samples were analysed in the same way as the ATOH1 transduced samples. We did not include a figure as it provides no more information than the number of myosin VIIa +ve cells that is given in the text and Figure 4.

How was the dose of TAPI chosen? How does the author decide that is the optimal dose for human utricles? The authors also discussed the use of DAPT (they said DAPI) and LY411575 and found no definitely effects. Thus, it is difficult to determine if Notch inhibition (or when combined with Atoh1) is less effective in inducing new hair cells or not.

The doses used are those reported in the paper by Lin et al., 2011 as we now mention in the Discussion section. We agree that we cannot be sure that Notch inhibition is less effective. We have added in the Discussion section:

“It may also be that higher concentrations of agents that inhibit the Notch pathway are required in the human tissue than in mouse and this may warrant further investigation”.

DAPI now corrected to DAPT.

With regards to SEM and TEM analyses in Figure 5, it is unclear how the authors determine that the pictures represent the GFP-transfected cells without co-labeling. As stated earlier, the information on the rate and cell types transfected and also health of harvested tissues will help the readers determine the meaning of these TEM and SEM analyses.

The reviewer is correct that SEM and TEM analyses do not directly show that cells are myosin VIIa +ve and that we would need to perform immunogold labelling to show this. We are working on that. We do not state specifically that the particular cells shown in EM images are transduced cells expressing GFP. What we say is that in tissue that had been transduced, SEM shows numerous cells that bear clusters of microvilli which are not seen in control tissue (figure now included Figure 2I), At the very least this observation shows that ATOH1 transduction induced some kind of phenotypic alteration to supporting cells. Furthermore, transduced immunolabelled tissue examined as whole mounts showed cells with clusters of microvilli the same as those seen by SEM were GFP+ve and myosin VIIa +ve (Figure 5E). Additionally, when after confocal examination of individual samples of the immunolabelled tissue, those same samples were subsequently prepared for and examined by SEM, numerous cells bearing clusters of microvilli were evident. Likewise, when samples examined by SEM that showed numerous cells bearing clusters of microvilli were subsequently prepared for thin sections and examined by TEM, distinct cells that bore clusters of elongated microvilli showed features reminiscent of hair cells. Taken together it would seem to be reasonable to infer that the transduction led to the generation of cells with clusters of microvilli that had some morphological features consistent with that of hair cells.

Lastly, the RNAseq experiments will benefit from further validation of genes up- and downregulated.

Clarified.

Reviewer #3:This is an important paper that provides evidence for regeneration of hair cells in the vestibular epithelium in aged human tissue. This reviewer appreciates the rarity of being able to analyse such tissue and the authors have done an excellent job in getting the most out of this valuable tissue.Discussion section: "the generated cells are sometimes seeming in contact with each other" Do the authors have any transmission electron microscopy data? If so, it would be good to include it, but of course, it is not necessary to do new experiments.

We do not have any TEM images of cells with hair cell-like features contacting each other.

Are the authors suggesting here that the newly-generated cells are derived from a single clone?

No. The point we were making here, to which we refer in the final paragraph of the Discussion section, is that while cells with some hair-cell like feature were generated, the normal regular mosaic whereby each hair cell is separated from its neighbours by intervening supporting cells is not re-created. This may have functional implications.

Subsection “Quantification”: For the cell counts, can the authors identify which region of the utricular macula the counts were taken? It would make a difference whether the analysis was done in the striolar (central) region versus the extrastriolar (peripheral) region, as the striolar region is damaged first and develops first during normal development. I think this would be an important thing to analyse if the confocal stacks can still be visualised.

See response to reviewer 1.

Figure 6B: This reviewer is not certain that the apoptotic body is inside a supporting cell, not unless the supporting cell is also dying. The cytoplasm of the cell in question, although it appears to be touching the basement membrane, is much lighter than the surrounding cells, which are indeed supporting cells, as one can tell by the presence of secretory granules within their cytoplasm and their more clumped heterochromatin. Although it seems unlikely that one hair cell would endocytose another, stranger things have happened during cell death.

This is Figure 2B. We have replaced the original panel with an alternative from another sample which shows the same things and hopefully is less contentious.

Figure 6F: Are the rod-like inclusions the same as what have been termed "cytocauds"? Can the authors speculate upon whether these are re-absorbed hair bundles?

This is Figure 2F In the text we already state:

“The hair cell bodies that persisted for up to 8 days after gentamicin exposure […] were always rounded in shape and most contained an actin-rich rod-like inclusion structure (Figure 2E, F, G) indicative of pathology being reminiscent of a cytocaud observed in damaged guinea pig vestibular hair cells (Kanzaki et al., 2002), and in mice appearing in damaged hair cells destined for phagocytosis by supporting cells (Bucks et al., 2017)”.

The latter paper suggests that these actin inclusions somehow play a role in the active uptake of the damaged hair cell by the supporting cell. Thus, they may be an indication of a dying hair cell destined for removal. We do not know their origin, but they do label for the stereociliary protein espin, though we have not mentioned this in the paper, as we do not believe it is directly relevant to the theme of this paper.

[Editors' note: further revisions were requested prior to acceptance, as described below.]

We felt that you had addressed comments from reviewers 1 and 3 satisfactorily, but reviewer 2 had asked for further validation of your RNAseq data. In your response, you stated that this had been 'clarified'. However, this did not actually seem to have been addressed anywhere. Suitable experiments would include qPCR and/or in situ hybridisation or immunohistochemistry for a selection of the genes that you have identified via RNASeq as up- or down-regulated after ATOH1 mis-expression. This additional validation was listed as an essential revision in the initial decision letter, and this requirement still stands. Please see the detailed comments from the reviewer below.Reviewer #2:The authors have done a decent job addressing the reviewers' concerns to improve the candidacy of this manuscript. Specifically, the methods used to quantify hair cells and the variability of hair cell number are described and the use of a supratherapeutic dose of aminoglycoside to ablate "all" hair cells explained. However, it is still somewhat perplexing that authors think that hair cells without bundles can take up aminoglycosides and be ablated.Nonetheless, more convincing pictures of Atoh1-GFP, Myosin7+ hair cells were shown, indicating that supporting cells are competent.

We are pleased that we have addressed satisfactorily the comments from reviewers 1 and 3 and “have done a decent job addressing [most of reviewer #2’s'] concerns to improve the candidacy of this manuscript”. However, reviewer #2 and the editors both ask for some means to “validate” the RNA sequencing data as an essential revision. In line with the editors’ suggestions that “suitable experiments would include qPCR […] for a selection of the genes […] identified via RNASeq as up- or down-regulated after ATOH1 mis-expression”, we have now performed such experiments. These have confirmed the same change in direction and similar expression level for 5 genes – listed in Table 1 in the paper- that RNA-sequencing showed to be either up (3 genes) or down (2 genes) regulated after transduction with ATOH1. We have amended the text accordingly.

In subsection “Gene expression changes with Ad2-ATOH1-GFP transduction”:

“To validate results from the RNA-sequencing, quantitative RT-PCR was conducted on five of the genes listed in Table 1. These were CITED, IRF9, SNAI1, EP300 and HDAC9. In all cases the trends for fold-changes were the same as in RNA-seq and in most cases they were very close in actual values (Figure 6).”

We have added the data from that analysis as an additional figure (Figure 6).

And in subsection “Quantitative RT-PCR”:

“Taqman assays for CITED, IRF9, SNAI1, EP300 and HDAC9 (Hs 00388363, Hs00196051, Hs00195591, Hs00914223 and Hs^-1^081558) were purchased from ABI and were run in technical triplicates on an ABI QuantStudio Real Time PCR System under manufacturer’s standard parameters for comparative C_T_ analysis. Total polyA+ RNA from two ATOH1 utricle transduction experiments (5 days post gentamicin) were separately converted into cDNA. These were then pooled and constituted the ATOH samples. Total polyA+ RNA from three control utricle samples (2 days, 8 days and 14 days post gentamicin) were separately converted into cDNA. These were then pooled and constituted the control samples. Amplifications were normalized to a GAPDH internal control (ABI Taqman assay Hs99999905).”

However, Figure 3D still shows that both actin and DAPI in blue (I think this is a typo). Lastly, it is stated that RNAseq experiment results should be validated, this was not addressed even though the author states "clarified". Since mechanism of Atoh1 can be viewed as a strength of this paper, validation of these data will help its candidacy.

This is not a typo. Both labelling of actin with fluorescently-tagged phalloidin and DAPI labelling of nuclei are in the blue channel. The labelling for actin is shown to provide orientation and to show the integrity of the tissue. The legend has been amended to include:

“Phalloidin-labelled actin, as well as DAPI to label nuclei, is in the blue channel to label the intercellular junctions as orientation for identification of the luminal surface of the epithelium”.

We hope we have now addressed satisfactorily all the outstanding concerns of reviewers and editors and that you will now find the paper acceptable for publication in *eLife*.